# Evolutionary repair: Changes in multiple functional modules allow meiotic cohesin to support mitosis

**Yu-Ying Phoebe Hsieh**[1]*, **Vasso Makrantoni**[2], **Daniel Robertson**[2], **Adèle L. Marston**[2], **Andrew W. Murray**[1]*

**1** Department of Molecular and Cellular Biology, Harvard University, Cambridge, Massachusetts, United States of America, **2** The Wellcome Centre for Cell Biology, University of Edinburgh, Edinburgh, United Kingdom

* phoebohsieh@gmail.com (Y-YPH); awm@mcb.harvard.edu (AWM)

**Data Availability Statement:** All non-WGS data are within the paper and its Supporting Information files. WGS data are held in public repositories: the NCBI Gene Expression Omnibus under the

## Abstract

The role of proteins often changes during evolution, but we do not know how cells adapt when a protein is asked to participate in a different biological function. We forced the budding yeast, *Saccharomyces cerevisiae*, to use the meiosis-specific kleisin, recombination 8 (Rec8), during the mitotic cell cycle, instead of its paralog, Scc1. This perturbation impairs sister chromosome linkage, advances the timing of genome replication, and reduces reproductive fitness by 45%. We evolved 15 parallel populations for 1,750 generations, substantially increasing their fitness, and analyzed the genotypes and phenotypes of the evolved cells. Only one population contained a mutation in Rec8, but many populations had mutations in the transcriptional mediator complex, cohesin-related genes, and cell cycle regulators that induce S phase. These mutations improve sister chromosome cohesion and delay genome replication in Rec8-expressing cells. We conclude that changes in known and novel partners allow cells to use an existing protein to participate in new biological functions.

## Introduction

Conserved biological functions, such as respiration, DNA replication, and chromosome segregation, are the result of the concerted action of many proteins. Although individual proteins have defined biochemical functions, such as those of kinases, proteases, and filament-forming proteins, the role they play in biological processes is determined both by their own biochemical properties and the interactions of these properties, both direct and indirect, with those of other molecules that contribute to the biological process. As a result, different members of a protein family can participate in different but related processes, such as the three different cyclin-dependent kinases (Cdks) that induce exit from G1 (Cdk4 and Cdk6), DNA synthesis (Cdk2), and mitosis (Cdk1) in the mammalian cell cycle [1,2].

How does evolution alter the properties of a protein and its interaction with other partners to allow it to participate in different biological functions? One approach to answer this question is to use experimental evolution to ask what must change, either in the candidate protein

accession number GSE 141598 and the NCBI
BioProject under the accession number
PRJNA594153.

**Funding:** Funding sources for authors include
Wellcome Senior Fellowship 107827 to ALM,
https://wellcome.ac.uk/funding/schemes/senior-
research-fellowships; Welcome Centre core grant
203149 to ALM, https://wellcome.ac.uk/; National
Institute of Health RO1-GM43987 to AWM, https://
projectreporter.nih.gov/project_info_description.
cfm?aid=9600703&icde=48408323; and National
Science Foundation-Simons Center for the
Mathematical and Statistical Analysis of Biology,
#1764269 (NSF) and #594596 (Simons) to AWM,
https://www.nsf.gov/awardsearch/showAward?
AWD_ID=1764269&HistoricalAwards=false. The
funders had no role in study design, data collection
and analysis, decision to publish, or preparation of
the manuscript.

**Competing interests:** The authors have declared
that no competing interests exist.

**Abbreviations:** Cdk, cyclin-dependent kinase;
Cdk8, cyclin-dependent kinase 8; ChIP, chromatin
immunoprecipitation; Clb, cyclin B; DDK, Dbf4-
dependent kinase; dNTP, deoxyribonucleotide
triphosphate; esp1-1, extra spindle pole bodies
1–1; GAL1, galactose metabolism 1; GFP, green
fluorescent protein; HA, hemagglutinin; HU,
hydroxyurea; Hxk1, hexokinase; Mad2, mitotic-
arrest deficient 2; ProA, proline mutant A; qPCR,
quantitative polymerase chain reaction; Rec8,
recombination 8; SMC, structural maintenance
complex.

or elsewhere in the genome, to allow one protein to perform the function of another, related protein. We applied this approach to the kleisin protein family, whose members organize the structure of chromosomal DNA. In prokaryotes and eukaryotes, kleisins bind to structural maintenance complex (SMC) proteins to form a proteinaceous ring [3] that interacts with chromosomes. In most bacteria and archaea, there is a single kleisin and SMC protein [4,5]. In eukaryotes, kleisin and SMC proteins have duplicated and acquired specialized functions [3,6]. Kleisin-γ proteins associate with Smc2/Smc4 heterodimers to form the condensin complex [7], which regulates chromosome structure in mitosis and meiosis. Kleisin-α proteins interact with Smc1/Smc3 heterodimers to form cohesin, the complex that holds sister chromosomes together [8] and regulates the timing of chromosome segregation: cohesin holds chromosomes together from S phase, when DNA replication occurs, until the proteolytic cleavage of kleisin by separase opens the ring and allows sister chromosomes to separate from each other.

Most eukaryotes have two different kleisin-α proteins. The mitotic kleisin holds sister chromosomes together in mitosis, whereas the meiotic kleisin is expressed only in meiosis [9]. Both kleisins interact with Smc1 and Smc3, but their proteolysis is regulated differently to produce the different patterns of chromosome segregation in mitosis and meiosis. In mitosis, the protease that cleaves kleisin acts uniformly on all parts of chromosomes, leading to sister chromosome separation [10–12]. In meiosis, however, the regulation of kleisin cleavage is modified to allow two rounds of chromosome segregation to follow a single round of DNA replication, producing four haploid genomes: in meiosis I, cleaving the kleisin on the chromosome arms allows homologous chromosomes to segregate from each other, and then in meiosis II, cleaving the remaining kleisin, located near the centromeres, allows sister centromeres to segregate from each other [10,13]. Both cohesin complexes participate in additional functions. Mitotic cohesin regulates chromosome condensation [14,15], gene expression [16], and DNA damage repair [17,18] and meiotic cohesin regulates meiotic recombination [19,20] and chromosome topology [21]. Most eukaryotes have both mitotic and meiotic kleisins, suggesting that the duplication and divergence of these paralogous proteins occurred at or soon after the evolutionary origin of eukaryotes [22–24]. Both mitotic and meiotic kleisins retain their interaction with Smc1 and Smc3 and their role as targets of cell cycle–regulated proteolysis, but they interact differently with other partners to allow them to play distinct roles in the different patterns of chromosome segregation in mitosis and meiosis.

Despite their conserved biochemical function, the mitotic kleisin has difficulty in supporting the biological function of meiotic chromosome segregation and vice versa. In the budding yeast, *Saccharomyces cerevisiae*, the mitotic kleisin is encoded by *SCC1* and the meiotic kleisin is encoded by recombination 8 (*REC8*). Replacing the coding sequence of *REC8* by that of *SCC1* during meiosis disrupts meiotic chromosome segregation [20,25] and the opposite experiment, expressing *REC8* from the *SCC1* promoter in mitosis, slows cell proliferation [13]. These results suggest that the functional difference between yeast kleisin proteins is mainly determined by the difference between their amino acid sequences rather than the promoters that control their expression. The mitotic and meiotic kleisins are thus an example of a common theme in protein evolution: an ancient separation between the function of two paralogous proteins that participate in different biological processes using biochemically similar functions. By substituting the meiotic kleisin for its mitotic relative, we can reveal what defects this replacement causes and ask how rapidly evolution can repair them. The defects help us to understand how the evolution of a protein's biochemical function makes it specific for a given biological process. The mutations that are selected to repair the defects reveal genes that interact functionally with kleisin and the relative importance of mutations in kleisin and in its interacting partners in changing its role in a biological function.

Substituting the budding yeast meiotic kleisin, Rec8, for its mitotic counterpart, Scc1, reduces the reproductive fitness of budding yeast by 45%. We evolved parallel yeast populations that expressed Rec8 in place of Scc1 for 1,750 mitotic generations. Only one population had a mutation in Rec8, but we repeatedly found adaptive mutations in the transcriptional mediator complex, cell cycle regulators that induce the G1-to-S transition, and cohesin-related genes; these mutations restore sister chromosome cohesion and thus increase the fitness of the evolved populations. Unexpectedly, we found that replacing Scc1 with Rec8 leads to earlier firing of replication origins. All three classes of adaptive mutations restored the timing of genome replication to the wild-type pattern. Engineering mutations that reduced replication origin firing or slowed replication forks improved the fitness of Rec8-dependent cells, revealing a new link between genome replication and sister chromosome cohesion. Our results suggest that cells can adapt rapidly to use a protein in a different biological function by modifying the partners that the protein interacts with, directly or indirectly, rather than modifying the protein that must participate in the new function.

## Results

### Using the meiotic kleisin, Rec8, for mitosis leads to multiple defects

We examined the consequences of replacing Scc1 with Rec8 in the mitotic cell cycle (Fig 1A). Previous studies showed that replacing Scc1 with Rec8 impairs mitotic growth [13] and DNA damage repair [17], showing that Rec8 cannot completely substitute for Scc1 in mitosis. We compared the reproductive fitness and the cellular and molecular phenotypes of the Rec8- and Scc1-expressing strains, referring to the Scc1-expressing strain as wild type. We used competitive growth to measure the fitness of the Rec8-expressing strain relative to wild type in rich media: the fitness of the Rec8-expressing strain is only 55% that of wild type (Fig 1B).

We examined sister chromosome cohesion and chromosome segregation in Rec8-expressing cells. Because cells mis-segregating chromosomes become progressively more aneuploid, we wanted to examine acute rather than chronic effects of replacing Scc1 with Rec8. We expressed *REC8* from the endogenous *SCC1* promoter and conditionally expressed an additional copy of *SCC1* from the galactose metabolism 1 (*GAL1*) promoter. The *GAL1* promoter is rapidly repressed by glucose [26,27], allowing us to repress *SCC1* expression rapidly and study the function of Rec8 in a single mitotic cell cycle. We confirmed that when *SCC1* is turned off, expressing Rec8 slows progress through the cell cycle (S1A Fig). We assayed cohesion between sister chromosomes by following a single, green fluorescent protein (GFP)-tagged chromosome through mitosis. Chromosome 5 was labeled by the binding of a GFP-*tet* repressor fusion to an array of *tet* operators integrated near the centromere [28,29]. We asked if Rec8 could hold sister chromosomes together from S phase to mitosis by following the GFP-labeled centromeres (henceforth GFP dots) under the microscope as cells were released from a G1 arrest, allowed to proceed synchronously through the cell cycle, and then arrested in mitosis by benomyl that depolymerizes microtubules. In this assay, a pair of linked sister chromosomes appears as a single GFP dot, whereas sister chromosomes that have lost cohesion appear as two GFP dots (Fig 1C). The fraction of cells with two GFP dots in a population indicates the degree to which sister chromosomes have separated. From S phase to mitosis, the majority of the wild-type population showed a single GFP dot as expected (Fig 1C). In the Rec8-expressing strain, 10% of the population showed two GFP dots during S phase and this fraction rose to 50% in mitosis (Fig 1C). During a single cell cycle, the defect in sister chromosome cohesion of the Rec8-expressing strain was smaller than the defect in a strain completely lacking Scc1 (Fig 1C), showing that Rec8 retains some cohesin function.

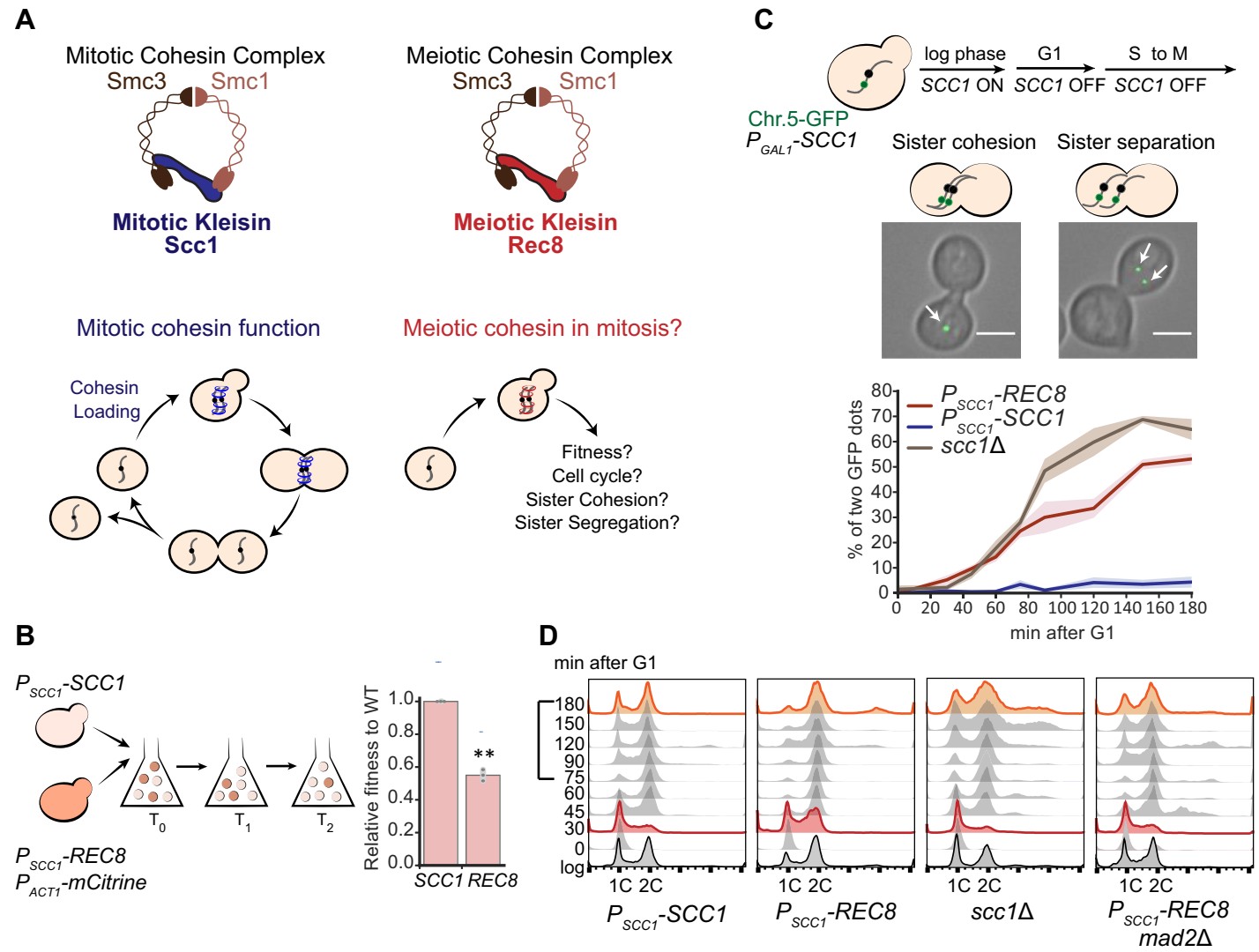

**Fig 1. Expressing Rec8 in place of Scc1 impairs the mitotic cell cycle and sister chromosome cohesion. (A)** A diagram of the mitotic and meiotic cohesin complexes. Mitotic cohesin holds replicated sister chromosomes together in mitosis. We investigated the ability of the meiotic cohesin to support mitosis. **(B)** The fitness of Rec8-expressing cells is 55% of a wild-type strain expressing Scc1. The Rec8-expressing strain expressed a fluorescent marker ($P_{ACT1}$-mCitrine) and was competed against wild type. The fitness of Rec8-expressing cells relative to wild type was calculated as changes in the ratio of these two strains over multiple generations. In the right panel, the darker gray points represent the values of three biological replicates, and the thinner gray bar represents one standard deviation on each side of the mean of these measurements. (two-tailed Student $t$ test, $^{**}p < 0.01$). **(C)** The Rec8-expressing strain cannot maintain sister chromosome cohesion in mitosis. All the strains ($P_{SCC1}$-REC8, $P_{SCC1}$-SCC1, and scc1Δ) carried a $P_{GAL1}$-SCC1 copy integrated in the genome to allow the acute effect of altered kleisin expression to be analyzed. To examine sister chromosome cohesion in a single cell cycle, Scc1 expression was switched off in G1-arrested populations by transferring cells to YEP containing 2% raffinose and α-factor. Cells were released to YPD containing benomyl to resume cell cycle and held in mitosis. Two different patterns of sister chromosome cohesion are shown: A budded cell with a single GFP dot represents functional sister chromosome cohesion; a budded cell with two GFP dots represents lack of sister chromosome cohesion. A single GFP dot is marked with a white arrow. At least 100 cells were imaged at each time point in each experiment. Three biological repeats were performed at each time point and for each strain; the right panel showed mean and standard deviation for the wild type (blue), Rec8-expressing (red), and scc1Δ (brown) strains. The scale bar is 5 μm. Data associated with this figure can be found in S1 Data. **(D)** The Rec8-expressing strain progressed through S phase faster and through mitosis slower. All strains were cultured as in Fig 1C, but cells were released in YPD to allow completion of the first cell cycle and entry into the second. Samples were collected at the indicated time points to examine DNA content by flow cytometry. Cell cycle profiles at 30 and 180 minutes are labeled in red and orange, respectively. For the cells expressing Scc1, cells are predominantly in S phase at 30 minutes, predominantly in G2 and mitosis from 45 to 75 minutes, and return to G1 at 90 minutes. Cells that express Rec8 have a long delay in mitosis, which is abolished by removing the spindle checkpoint. For examining the progression through mitosis, the timepoints from 75 to 180 minutes are highlighted by a square bracket on the y-axis. Chr.5-GFP, Chromosome 5-green fluorescent protein; GFP, green fluorescent protein; Rec8, recombination 8; Scc1, sister chromosome cohesion 1; Smc, structural maintenance complex; YEP, yeast extract and peptone; WT, wild type; YPD, yeast extract, peptone, and dextrose.

Sister chromosomes must be linked to each other to allow their kinetochores to attach stably to opposite poles of the mitotic spindle. In this orientation, forces exerted on the kinetochores create tension that inactivates the spindle checkpoint, leading to the activation of anaphase-promoting complex, the activation of separase, the cleavage of kleisin, and the onset of anaphase [10]. The Rec8-expressing strain frequently failed to correctly orient sister kinetochores (S2 Fig), a defect likely to cause errors in chromosome segregation (S1B Fig). We hypothesized that the sister chromosome cohesion defect would activate the spindle checkpoint, thus prolonging mitosis. By tracking a G1-synchronized population, we found that cells with a 2C DNA content (which are in G2 or mitosis) persisted for longer in the Rec8-expressing strain than they did in wild type (compare the time points from 75 to 180 minutes, Fig 1D). Removing mitotic-arrest deficient 2 (Mad2), a spindle checkpoint protein [30,31], from the Rec8-expressing strain, increases the fraction of cells with a 1C DNA content (from 75 to 180 minutes, Fig 1D), suggesting that the sister chromosome cohesion defect activates the spindle checkpoint. In addition, compared to wild type, the Rec8-expressing strain showed more cells with a 2C DNA content at 30 minutes after entering cell cycle. This phenotype is not due to faster escape from a G1 arrest, because budded cells accumulate with indistinguishable frequency in wild-type and Rec8-expressing strains (S3 Fig). Unexpectedly, the *mad2Δ*, Rec8-expressing strain progressed through S phase with similar kinetics to wild type (see Discussion). In cells lacking kleisin or the cohesin loading complex mutant, the progression of S phase is identical to wild type [29], suggesting that the altered S phase progression is a specific feature of Rec8-expressing cells. In summary, replacing Scc1 with Rec8 leads to profound defects in sister cohesion and alters genome replication by an unknown mechanism.

We asked whether the phenotype of Rec8-expressing cells could be explained by reduced cohesin levels or reduced cohesin binding to mitotic chromosomes. We measured Rec8 levels in a synchronous cell cycle and compared them with those of Scc1 (Fig 2A). Scc1 was barely detectable in G1, peaked during S phase, and its cleavage product was detected at 60 minutes as cells entered anaphase. During G1 and S phase, there was 4-fold less Rec8 than Scc1. At 90 minutes, the Rec8 protein level decreased but we did not detect the cleavage product of Rec8, either because the onset of anaphase was asynchronous or the cleavage product is too unstable to be detected [13]. The lower Rec8 protein level could be due to inefficient protein synthesis or protein instability. We tested the second hypothesis by examining the stability of Scc1 and Rec8 in mitotically arrested cells: the half-life of Scc1 is 200 minutes, whereas Rec8 has a half-life of only 58 minutes (Fig 2B). The instability of Rec8 is due to the weak separase activity that exists outside anaphase [11]: reducing separase activity with a temperature sensitive mutation, extra spindle pole bodies 1–1 (*esp1-1*) [32], increases the half-life of Rec8 to 190 minutes.

Finally, we compared the binding of Rec8 and Scc1 to chromosomes by chromatin immunoprecipitation (ChIP). In mitotically arrested cells, immunoprecipitating Rec8 brought down less DNA at canonical cohesin binding sites than Scc1 (Fig 2C). This was not only true for individual sites but was also observed genome-wide. Calibrated ChIP-Seq revealed reduced levels of chromosomal Rec8 compared to Scc1 at peri-centromeres, where cohesin is most enriched, across all 16 chromosomes (Figs 2D and S4A). Among all chromosomes, the enrichment of Rec8 specifically at core centromeres is higher compared to that of Scc1, but lower in the flanking peri-centromeres (Figs 2D and S4B). This suggests that Rec8-containing cohesin is less efficient than Scc1 in translocating from its loading site at centromere. The reduced overall binding of Rec8 on mitotic chromosomes might be partially explained by the lower Rec8 level in mitosis (S5A Fig). To test this idea, we asked whether equivalent amounts of ectopically produced Rec8 and Scc1 can be loaded on chromosomes in G1. Although Scc1 is not normally expressed until the onset of S phase, ectopically expressing Scc1 in G1 allows cohesin to be loaded on chromosomes [34]. We expressed Rec8 or Scc1 from the *GAL1* promoter in

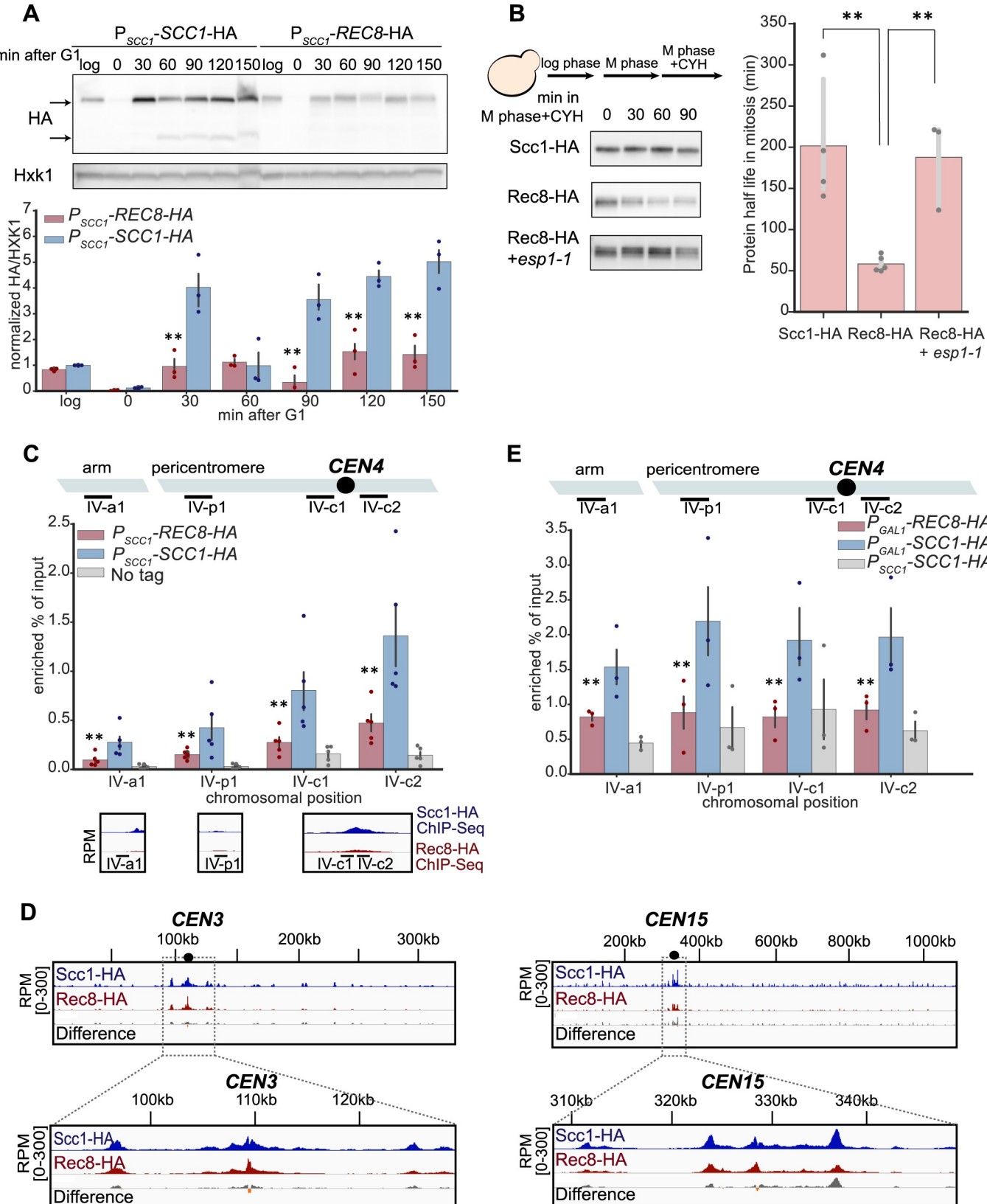

**Fig 2. Rec8 is unstable and shows reduced binding to mitotic chromosomes. (A)** The Rec8 protein level is lower than Scc1 in mitosis. Both *SCC1* and *REC8* were expressed from the *SCC1* promoter and fused to a triple hemagglutinin tag (3xHA) at their C termini in a strain that also carried $P_{GAL1}$-*SCC1*. To follow kleisin protein levels in a single mitotic cycle, expression from the *GAL1* promoter was repressed and cells were released from a G1 arrest and allowed to proceed through the cell cycle as in Fig 1D. Cells were collected at the indicated time points, and cell extracts were obtained by alkaline lysis prior to analysis by western blotting. Hxk1 was used as a loading control. In the blot, the upper arrow marks the size of full-length protein and the bottom arrow marks the size of cleavage product. The three colored points represent the values of three biological replicates, and the dark gray bar represents one standard deviation on each side of the mean of these measurements. The statistical significance between data from the Rec8-expressing strain and wild type was calculated by two-tailed Student *t* test, $^{**}p < 0.01$. Raw images associated with Fig 2A can be found in S1 Raw Image. Data associated with this figure can be found in S1 Data. **(B)** The instability of Rec8 in mitosis depends on separase activity. Cells were grown to log phase in YPD at 30 ˚C and held in mitosis by addition of benomyl. To check protein stability, cycloheximide was added to the cultures to inhibit protein synthesis. Cells were collected every 30 minutes to examine protein levels by western blotting. Both Scc1 and Rec8 were detected by an anti-HA antibody. The darker gray points represent the values of at least three biological replicates, and the thinner gray bar represents one standard deviation on each side of the mean of these measurements. The half-life of Rec8 is increased in cells expressing a temperature-sensitive mutant of Esp1 (*esp1-1*) (two-tailed Student *t* test, $^{**}p < 0.01$). Raw images associated with Fig 2B can be found in S1 Raw Image. Data associated with this figure can be found in S1 Data. **(C)** The level of chromosome-bound Rec8 is lower than that of Scc1 in mitosis. Strains were released from a G1 arrest, proceeded synchronously through one cell cycle, and then were arrested in mitosis in YPD containing benomyl. The chromosome-bound kleisin protein was immunoprecipitated using an anti-HA antibody. Chromatin lysates were prepared from wild type, a Rec8-expressing strain, and a strain without HA tag as negative control. The level of chromosome bound kleisin at the known cohesin binding sites was measured by the amount of DNA that associated with the immunoprecipitated kleisin. DNA was measured by qPCR and expressed as the fraction of material compared to the total chromatin lysate (shown in the y-axis). Four genomic loci on Chromosome 4 are shown. The colored points represent the values of five biological replicates and the dark gray bar represents one standard deviation on each side of the mean of these measurements. The statistical significance between data from the Rec8 strain and wild type was calculated by two-tailed Student *t* test, $^{**}p < 0.01$. The bottom panel shows the ChIP-Seq data of Scc1 and Rec8 at the corresponding cohesin binding sites, under the same conditions. Data associated with this figure can be found in S1 Data. **(D)** ChIP-Seq analysis of Scc1 and Rec8 binding in mitosis. Sample preparation and immunoprecipitation were done as in Fig 2C. The immunoprecipitated DNA bound by Scc1 or Rec8 were examined by whole genome sequencing. The amount of immunoprecipitated DNA is expressed as reads per million (RPM) calibrated to the reference, *Schizosaccharomyces pombe* genomic DNA immunoprecipitated by the Scc1 ortholog, Rad21. The calibrated signal of ChIP-Seq data (relative to a control *S. pombe* sample) representing the degree of enrichment of kleisin (Scc1 in blue and Rec8 in red) and the difference in enrichment between the two kleisins (gray: Scc1's signal is more than Rec8's; orange: Rec8's signal is more than Scc1's) is visualized by the Integrated Genomic Viewer [33]. Chromosomes 3 and 15 are shown as examples of chromosomes with different sizes and different degrees of peri-centromeric Rec8 binding, with an expanded view of 20 kb DNA on each side of a centromere in the bottom panel. **(E)** Rec8 loads poorly on G1 chromosomes compared to Scc1. To overexpress kleisins in G1, the $P_{GAL1}$-*SCC1*-HA and $P_{GAL1}$-*REC8*-HA strains were arrested in YEP containing 2% galactose and α-factor. ChIP and qPCR were performed as described in Fig 2C. The darker points represent the values of three biological replicates and the darker gray bar represents one standard deviation on each side of the mean of these measurements. The statistical significance between data from the Rec8-expressing strain and wild type was calculated by two-tailed Student *t* test, $^{**}p < 0.01$. Data associated with this figure can be found in S1 Data. *CEN*, centromere; ChIP-Seq, chromatin immunoprecipitation sequencing; CYH, cycloheximide; Esp1, extra spindle pole bodies 1; *GAL1*, galactose metabolism 1; HA, hemagglutinin; Hxk1, hexokinase; qPCR, quantitative polymerase chain reaction; Rad21, radiation sensitive 1; Rec8, recombination 8; Scc1, sister chromosome cohesion 1; YEP, yeast extract and peptone; YPD, yeast extract, peptone, and dextrose.

G1-arrested cells and measured their chromatin association by ChIP–quantitative polymerase chain reaction (qPCR). Although the levels of ectopically produced Rec8 were slightly elevated compared to that of Scc1 (S5B Fig), it showed reduced accumulation on canonical cohesin binding sites (Fig 2E). We conclude that Rec8 is both less stable and, independently, associates less well with chromosomes compared to Scc1. We suggest that these molecular defects lead to defective sister cohesion, errors in chromosome segregation, and slower passage through mitosis, leading to the production of dead and aneuploid cells, thereby reducing the fitness of Rec8-expressing cells.

## Experimental evolution increases the fitness of Rec8-expressing strains

To study how cells adapt to a protein that performs an essential function poorly, we asked if experimental evolution would allow Rec8-expressing cells to acquire mutations that would improve their fitness; these mutations could occur either in *REC8* or elsewhere in the yeast genome. We constructed fifteen ancestral clones, each containing a deletion of the chromosomal *SCC1* gene and a centromeric plasmid expressing *REC8* from the *SCC1* promoter ($P_{SCC1}$-*REC8*). Ancestral clones were inoculated into rich media at 30˚C, and each culture was diluted 6,000-fold into fresh media once it reached saturation. This process was repeated, freezing samples every 125 generations, until the populations reached 1,750 generations (Fig 3A). At generation 375, the fitness of all the evolved populations had increased by 20%–30% relative to the Rec8-expressing ancestor. At the end of the experiment, the fitness of the

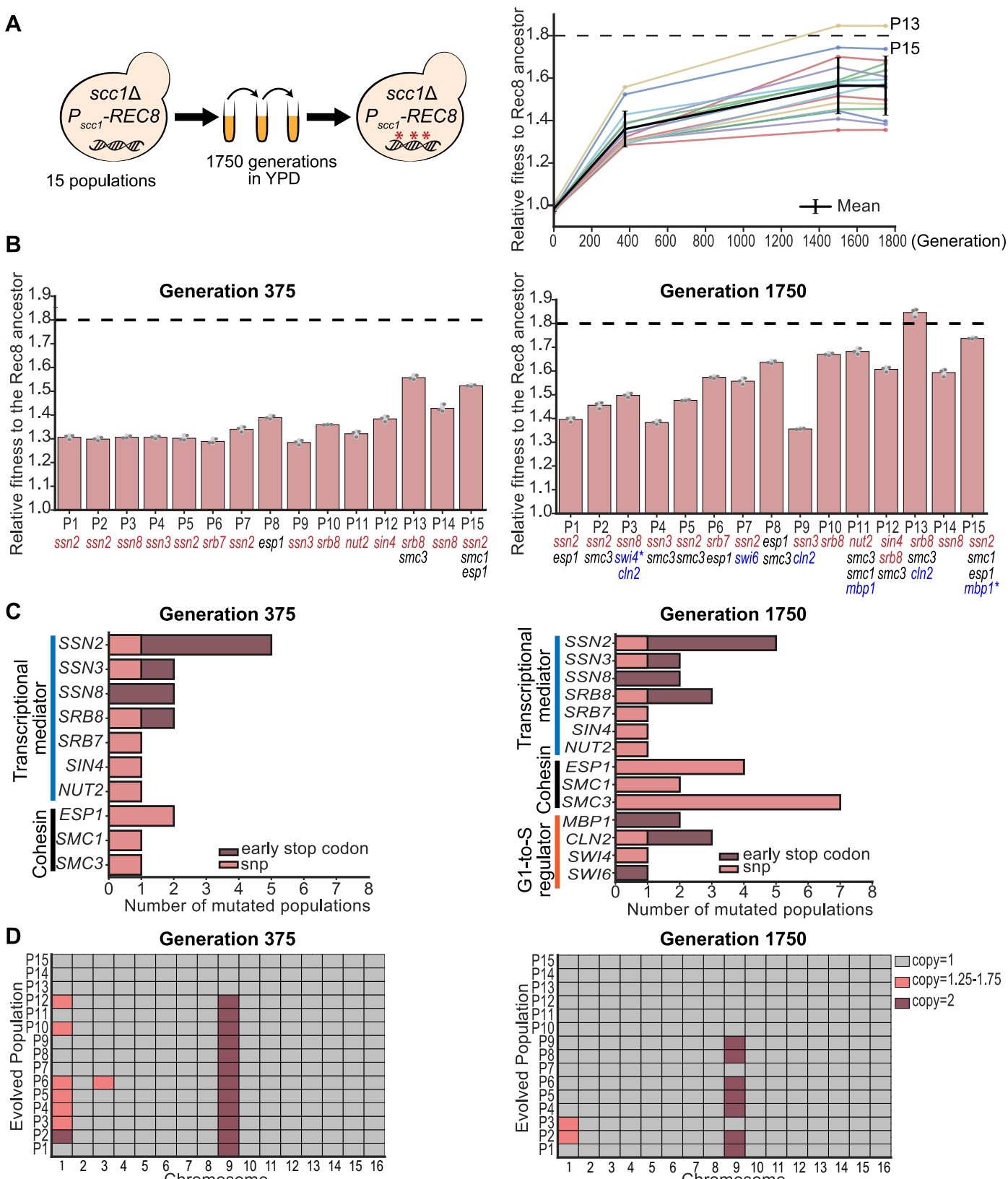

**Fig 3. Experimental evolution improves the fitness of Rec8-expressing populations. (A)** Schematic of the experimental evolution of 15 independent populations forced to use Rec8 in mitosis for 1,750 generations. All the evolved populations increased fitness during evolution (shown in the right panel). The mean relative fitness of each population during evolution is shown as an individual colored line. The average fitness of all 15 evolved populations is shown as a black line. The fitness of wild type relative to the Rec8-expressing ancestor is indicated as a black dashed line. Two evolved populations showing fitness near wild type are labeled P13 and P15. **(B)** The fitness of 15 evolved populations relative to the Rec8-expressing ancestor at generations 375 and 1,750. The relative fitness of each evolved population was measured by competing it against a fluorescently labeled ancestor in YPD. The darker gray points represent the values of three biological replicates, and the thinner gray bar represents one standard deviation on each side of the mean of these measurements. Genes in the three functional modules that mutate in at least six evolved populations at generation 1,750 are shown. Mutations in the different functional modules are color-coded: transcriptional mediator complex (red), G1-to-S cell cycle regulators (blue), and cohesin and its regulators (black). Genes mutated twice within the same population were marked with an asterisk (*). The black dashed line shows the fitness of wild type relative to the Rec8-expressing ancestor. The details of these mutations are available in S1 and S2 Tables. **(C)** Summary of functional modules that had acquired fixed mutations in more than six populations at generation 1,750. The x-axis shows the number of populations that acquire a mutation in any specified gene at generation 375 and 1,750. The y-axis shows mutated genes grouped by their functions: genes involved in the transcriptional mediator complex in blue, cohesin related genes in black, and the G1-to-S cell cycle regulators in orange. Mutations causing early stop codon are shown in dark red and single nucleotide changes are shown in pink. **(D)** Summary of changes in chromosomal copy number of all 15 evolved populations. The copy number of each chromosome was calculated by normalizing the median read depth of each chromosome to the median read depth over the entire genome. The results of 15 evolved populations at generations 375 and 1,750 are shown here: gray marks one copy, dark red marks two copies, and pink marks 1.25–1.75 copies, suggesting part of the population was disomic. The results of five ancestors are shown in S6A Fig. Data associated with Fig 3A–3D can be found in S1 Data. Rec8, recombination 8; YPD, yeast extract, peptone, and dextrose.

evolved populations was 30%–80% greater than that of the ancestor and the fitness of two evolved populations (P13 and P15) was similar to that of wild type (Fig 3A and 3B).

To identify adaptive mutations, we sequenced the genomes of five ancestral clones and pooled genomes of 15 evolved populations at generations 375 and 1,750. We focused on non-synonymous mutations that were present at a frequency ≥90% in any evolved population (S1 File). The evolved populations had an average of nine mutations at generation 375 and 17 mutations at generation 1,750 that met this criterion. We found multiple mutations in three functional modules: the transcriptional mediator complex, cohesin and its regulators, and regulators of cell cycle progression from G1 to S phase. At generation 375, 14 out of 15 evolved populations had a mutation in the transcriptional mediator complex, and four populations had mutations in the other two cohesin subunits, *SMC1* and *SMC3*, or separase, *ESP1* (Fig 3B and S1 Table). At generation 1,750, the early mediator mutations were still fixed, one population had acquired a mutation in a second mediator subunit (*SRB8*), and one population still lacked a mediator mutation (Fig 3B and S2 Table). Twelve out of the fifteen mediator mutations targeted the cyclin-dependent kinase 8 (Cdk8) complex, a regulatory module of mediator, and nine of these twelve mutations produced early stop codons (Fig 3C). Mutations in cohesin-related genes were common at generation 1,750: *SMC3*, *SMC1*, and *ESP1* were mutated in seven, two, and four evolved populations, respectively (Fig 3C). Seven populations had mutations in one of these genes, three populations had mutations in two genes, and five populations had not acquired mutations in any cohesin-related gene by generation 1,750. Four genes (*MBP1*, *CLN2*, *SWI6*, and *SWI4*) controlling the cell cycle transition from G1 to S were mutated in a total of six populations by generation 1,750, with both *CLN2* and *SWI4* mutated in population P3 (Fig 3B and 3C). In summary, nine out of 15 evolved populations acquired mutations both in the mediator complex and cohesin-related genes (Fig 3B). Only the three fittest populations had mutations in all three classes (cohesin-related, mediator, and G1-to-S regulators) and the fitness of two of these populations, P13 and P15, approached that of wild type (Fig 3B). We also searched for mutations in *REC8*, either in the coding sequence or the DNA 500 bp upstream and downstream of the ORF, using a lower threshold because our strains have two copies of *REC8*, one on the plasmid, driven by the *SCC1* promoter, and one on the chromosome driven by the *REC8* promoter. Population 12 had a single mutation (E148Q) on the plasmid-borne copy of *REC8*. Because this substitution is a conservative amino acid change, the position lies outside the regions of kleisin that are known to have interactions with other proteins, and it occurred in only one population, we do not know whether it is adaptive or a chance event.

In addition to point mutations, many evolved populations were aneuploid (Fig 3D). The five ancestral clones we sequenced had an extra copy of Chromosome 1, the smallest chromosome in budding yeast (S6A Fig). We think this reflects a combination of three factors, a very high frequency of chromosome mis-segregation in the ancestral Rec8-expressing cells, preferential mis-segregation of smaller chromosomes, and the small fitness cost of an extra copy of Chromosome 1 [35]. At generation 375, 12 populations had independently gained an extra copy of Chromosome 9, and seven also had an extra copy of Chromosome 1 or Chromosome 3 in part of the population. The other three were true haploids, with one having lost the extra copy of Chromosome 1 that was present in its ancestor. At generation 1,750, seven populations retained two copies of Chromosome 9, while the remaining populations had lost the extra Chromosome 9. Disomy for Chromosome 9 causes a slight fitness cost in wild type [35], but in our evolution experiment, the prevalence and persistence of Chromosome 9 disomes suggest that an extra copy of Chromosome 9 in Rec8-expressing cells is adaptive. We also detected two segmental duplication events at generation 1,750: population P4 duplicated part of Chromosome 4, and population P7 duplicated part of Chromosome 5 (S6B and S6C Fig). These duplicated regions are flanked by Ty1 transposons, suggesting segmental duplication is a result of recombination between homologous transposon sequences. The duplication on Chromosome 5 contains *SCC4*, whose duplication increases the fitness of strains that have cohesin defects that result from removing Ctf4 [36], and the other duplication lacks obvious candidate genes, making it unclear whether it was selected or a chance event.

The genetic alterations we found are specific to yeast cells adapting to expressing Rec8 rather than Scc1. Mutations in transcriptional mediator, cohesin-related genes, and the G1-to-S regulators have not been seen at frequencies that suggest they are adaptive in previous experimental evolution studies in *S. cerevisiae* [36–40]. In evolution experiments that improve the growth of haploid yeast in rich media, the most frequent ploidy change seen is diploidization [37,41], instead of gaining an extra copy of a specific chromosome, which has been seen for cells adapting to the absence of myosin [42] and growth at high temperature [43].

## Reconstruction confirms that candidate mutations are adaptive

We tested the effect of putative causative mutations by engineering them individually into the Rec8-expressing ancestor and examining the fitness and phenotypes of the resulting strains. We focused on four groups of genetic changes: mutations in transcriptional mediator, cohesin components, regulators of the G1-to-S transition, and an extra copy of Chromosome 9. Mutations in the transcriptional mediator complex primarily targeted the Cdk8 complex: of its four components, *SSN2* was mutated five times, *SSN3* and *SSN8* were each mutated twice, and *SRB8* was mutated three times. Nine out of these twelve mutations led to early stop codons, suggesting that these mutations inactivate the module's function. The mediator complex links the basic transcriptional machinery with transcription factors and controls various events in transcription, including transcriptional initiation, pausing, elongation, and the organization of chromatin structure [44]. The Cdk8 kinase module of mediator can positively or negatively regulate transcription [45]. We reconstructed mutations in three subunits (*SSN2*, *SSN3*, and *SSN8*) of the Cdk8 module. Each increased the fitness of the ancestor by 8%–25% (Fig 4A). Deleting the above genes also increased the fitness of the ancestor (Fig 4A), strongly suggesting that these evolved mutations are loss-of-function mutations. Of the mutations targeting cell cycle regulators, mutations in *MBP1* and two of three mutations in *CLN2* caused early stop codons (Fig 3C and S2 Table). We therefore mimicked the effect of these mutations by deleting the corresponding gene: *mbp1Δ* and *cln2Δ* increased the fitness of the ancestor by 20% and 25%, respectively (Fig 4B).

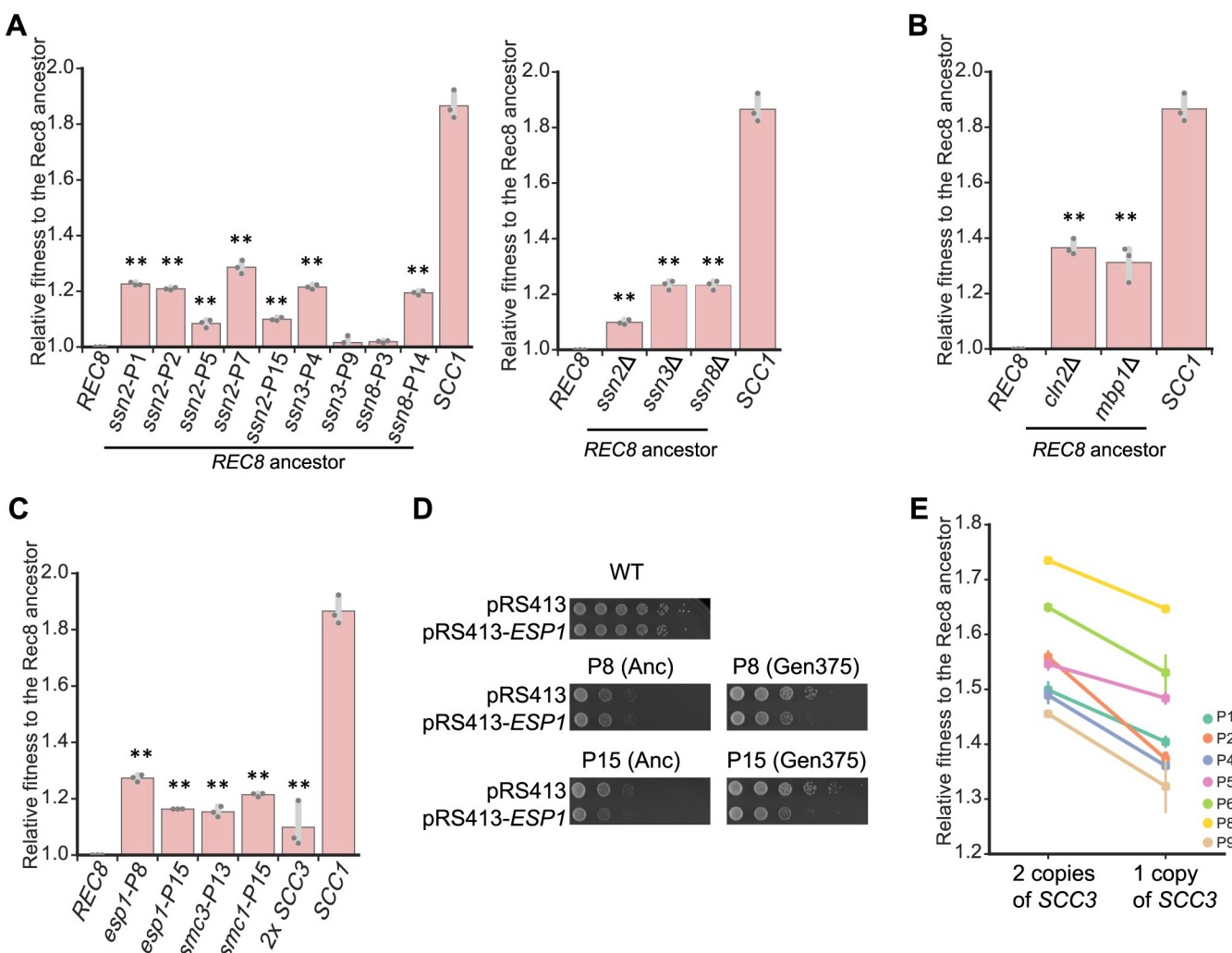

**Fig 4. Reconstructing individual evolved mutations increases the fitness of the Rec8-expressing ancestor. (A)** The effect of single evolved mutations and deletion of genes encoding subunits of the Cdk8 complex on fitness of the Rec8-expressing ancestor. **(B)** The effect of deleting *CLN2* and *MBP1* on fitness of the Rec8-expressing ancestor. **(C)** The effect of single evolved mutations in genes that encode other cohesin components or separase and an extra copy of *SCC3* on fitness of the Rec8-expressing ancestor. In **4A-4C**, each single evolved mutation was reconstructed in the Rec8 ancestral strain used in the evolution experiment. The relative fitness of reconstructed strains to the ancestor was measured by competing it against a fluorescently labeled Rec8 ancestor. The darker gray points represent the values of three biological replicates, and the thinner gray bar represents one standard deviation on each side of the mean of these measurements. The fitness of the wild-type strain, labeled as *SCC1*, is shown in each panel. The statistical significance between data from the Rec8-expressing strain and each mutation-reconstructed strain was calculated by two-tailed Student *t* test, ** $p < 0.01$. **(D)** *esp1* evolved mutations (*esp1-P8* and *esp1-P15*) are hypomorphic. A *CEN* plasmid carrying *ESP1* was transformed into a wild-type strain, two ancestors (Anc), and two evolved populations that had acquired *esp1* mutations (P8 and P15) at generation 375. Cells were subjected to 10-fold serial dilutions and spotted on YPD plates to assay growth. Cells transformed with an empty plasmid (pRS413) served as control. **(E)** The effect of deleting one copy of *SCC3* on fitness of the evolved populations with disomic Chromosome 9 at generation 1,750. Data associated with Fig 4A, 4B and 4C, and 4E can be found in S1 Data. Cdk8, cyclin-dependent kinase 8; *CEN*, centromere; *CLN2*, cyclin 2; *esp1*, extra spindle pole bodies 1; *MBP1*, mlul-binding protein; Rec8, recombination 8; SCC, sister chromosome cohesion; YPD, yeast extract, peptone, and dextrose.

The mutations in cohesin-related genes affect essential genes and are thus unlikely to eliminate the function of these genes. Individual mutations in *ESP1*, *SMC1*, and *SMC3* increased the fitness of the Rec8-expressing ancestor by 15%–31%, 14%, and 21%, respectively (Fig 4C). Our finding that Rec8 is sensitive to separase activity in mitosis (Fig 2B) raised the possibility that evolved *esp1* mutations are hypomorphic alleles that weaken separase activity. We tested this hypothesis by expressing an extra wild-type copy of *ESP1* in two evolved populations

carrying *esp1* mutations and their ancestors. As predicted, the extra copy of *ESP1* reduced the growth of these two evolved populations but not their ancestors (Fig 4D), suggesting that these evolved *esp1* mutations are hypomorphic. Consistent with this hypothesis, compromising separase activity by using a known temperature-sensitive mutation, *esp1-1* [32], increased the growth of Rec8-expressing cells at the permissive temperature (S7 Fig).

The prevalence of Chromosome 9 disomy in our evolved populations suggested that two copies of Chromosome 9 confer a selective advantage on Rec8-expressing strains. Aneuploidy has been adaptive in several evolution experiments by increasing the copy number of a specific gene [42,46]. Chromosome 9 encodes a candidate gene, *SCC3*, whose protein product pro-motes cohesin association with chromosomes [47] by interacting with the cohesin loading complex [48] and Scc1 [49]. We asked if an extra copy of *SCC3*, in the absence of the other genes on Chromosome 9, could increase the fitness of the Rec8-expressing ancestor. We inte-grated an extra copy of *SCC3* in the ancestor and found that this manipulation increased its fit-ness by 10% (Fig 4C), demonstrating that an extra copy of *SCC3* is sufficient to increase fitness. To test if an extra copy of *SCC3* is also necessary for increasing fitness, we deleted one copy of *SCC3* in clones from seven evolved populations that carried two copies of Chromo-some 9 at generation 1,750, reducing their fitness by 8% to 28% (Fig 4E). We conclude that an extra copy of *SCC3* explains much of the selective advantage of carrying an extra copy of Chro-mosome 9.

## Adaptive genetic changes restore sister chromosome cohesion

Do the adaptive mutations in Rec8-expressing strains increase fitness by improving sister chromosome cohesion? We engineered individual mutations into a Rec8-expressing strain containing a GFP-labeled Chromosome 5 and $P_{GAL1}$-*SCC1* and examined sister chromosome cohesion after acute depletion of Scc1. Individually deleting three subunits in the Cdk8 com-plex partially rescued the sister chromosome cohesion defect in cells where Rec8 was the only α-kleisin present (Figs 5A and S8). Amongst these genes, deleting *SSN3*, the kinase subunit of the Cdk8 complex, produced the greatest improvement in sister cohesion, comparable to the effect of the adaptive mutations in cohesin-related genes (*SMC1*, *SMC3*, or *ESP1*) or deleting the two genes that promote exit from G1, *CLN2* and *MBP1* (Fig 5A). An extra copy of *SCC3*, whose effect on fitness mimicked the Chromosome 9 disome, slightly improved sister cohesion (Fig 5A). Each evolved mutation also improved the accuracy of chromosome segregation in Rec8-expressing cells, with cohesin-related mutations having stronger effects than mediator mutations (S9 Fig). We conclude that mutations in transcriptional mediator, other cohesin components and separase, and cell cycle regulators can improve sister chromosome cohesion in Rec8-expressing cells.

We investigated the interactions between adaptive mutations in different functional mod-ules. The fitness of the evolved population P15 approached that of wild type at generation 1,750 and it had acquired mutations in four genes (*ssn2*, *esp1*, *smc1*, and *mbp1*) representing effects of mediator, separase, cohesin, and the G1-to-S transition. We investigated the interac-tion between these four mutations. To examine fitness and sister chromosome cohesion, we constructed double mutants in the strain carrying a GFP-labeled Chromosome 5 and $P_{GAL1}$-*SCC1*. For fitness, we saw two types of interactions: double mutations between any of *ssn2Δ*, *mbp1Δ*, and *smc1-P15* had a fitness that was indistinguishable from the sum of the effects of the individual mutations (Fig 5B–5D), whereas the *esp1-P15 ssn2Δ* and *esp1-P15 mbp1Δ* dou-ble mutants were substantially fitter than the sum of the fitness increases in the individual mutants (Fig 5E and 5F). For sister chromosome cohesion, all the double mutants had smaller defects in sister chromosome cohesion than either single mutant with the exception of the

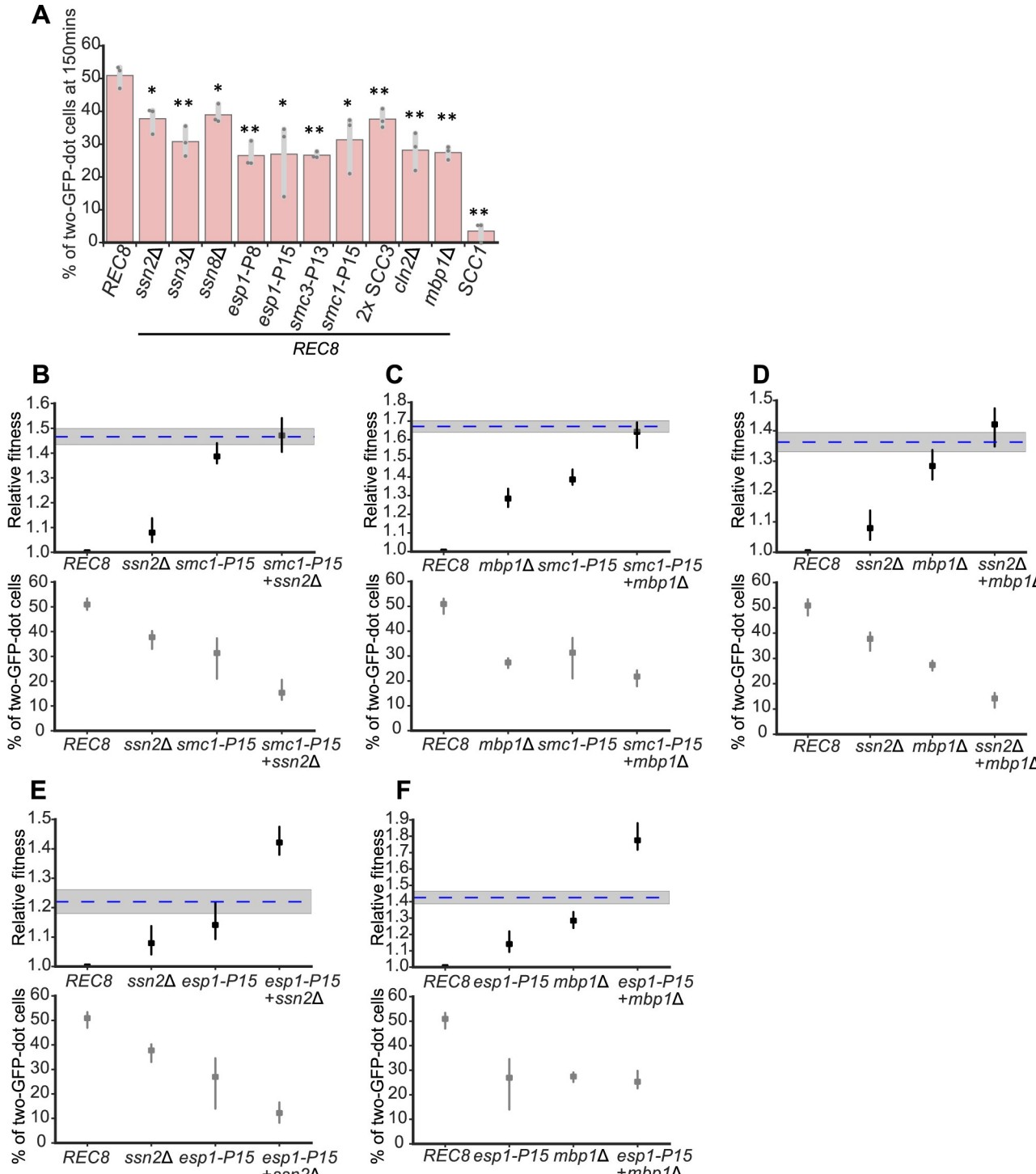

**Fig 5. Adaptive genetic changes improve sister chromosome cohesion in Rec8-expressing cells. (A)** Individual adaptive genetic changes partially improve sister chromosome cohesion. Deletions of genes in the Cdk8 complex, adaptive mutations in cohesin and its regulator, two copies of *SCC3*, and deletion of genes that regulate G1-to-S transition were reconstructed individually in the strain used for assaying sister chromosome cohesion. Cells were prepared as in Fig 1C, and the percentage of cells with two GFP dots in populations arrested in mitosis (150 minutes after releasing from G1) is shown. The darker gray points represent the values of three biological replicates, and the thinner gray bar represents one standard deviation on each side of the mean of these measurements. The statistical significance between data from the Rec8-expressing strain and each mutation-reconstructed strain was calculated by two-tailed Student *t* test, *$p < 0.05$, **$p < 0.01$. **(B-F)** Relative fitness and sister chromosome cohesion of double mutants are shown: *ssn2Δ* and *smc1-P15* **(B)**, *mbp1Δ* and *smc1-P15* **(C)**, *ssn2Δ* and *mbp1Δ* **(D)**, *ssn2Δ* and *esp1-P15* **(E)**, *mbp1Δ* and *esp1-P15* **(F)**. The blue dashed line represents

the expected fitness if two mutations contribute additively, and the shaded region represents the standard error of that expectation. Data associated with Fig 5A–5F can be found in S1 Data. Cdk8, cyclin dependent kinase 8; GFP, green fluorescent protein; Rec8, recombination 8; *SCC3*, sister chromosome cohesion 3.

*mbp1Δ esp1-P15* and *mbp1 smc1-P15* double mutants (Fig 5C and 5F), whose level of sister chromosome cohesion was either indistinguishable from or only slightly above that of the single mutants. This result suggests that *mbp1Δ* may have additional effects on fitness that are not mediated by improving sister chromosome cohesion. Overall, the interactions between mutations in different modules are additive or positively synergistic at the level of fitness and more complex at the level of sister cohesion.

We asked if the adaptive mutations altered the abundance of Rec8. We measured the Rec8 protein level in mitosis in seven strains, each containing an adaptive mutation in a different gene that appeared during our evolution experiment and was shown to increase the fitness of Rec8-expressing cells (S10 Fig). None of the mutations changed the level of Rec8, demonstrating that these adaptive mutations improve sister chromosome cohesion not by changing the amount of Rec8.

## Adaptive genetic changes delay the timing of genome replication and improve sister cohesion

Cell cycle progression profiles showed that the ancestral Rec8-expressing strain progressed through S phase faster than wild type (Fig 1D). Because the linkage between sister chromosomes is established in S phase [29] and all the adaptive mutations improved sister chromosome cohesion in the Rec8-expressing strain, we asked if these mutations also affected the dynamics of genome replication. By tracking cell cycle progression after release from a G1 arrest, we found deletion of three subunits in the Cdk8 complex and mutations in *SMC1*, *SMC3*, or *ESP1* decreased the fraction of cells that entered S phase right after exit from G1 (Fig 6A). We quantified the fraction of cells in S phase 30 minutes after release from a G1 arrest: 39% of wild-type cells were in S phase, whereas 57% of the Rec8-expressing cells were in S phase. In the Rec8-expressing strain, deleting genes encoding the subunits of the Cdk8 complex decreased the fraction of cells in S phase to 33%–50%, and mutations in *SMC1*, *SMC3*, and *ESP1* decreased this fraction to 17%–40% (Figs 6B and S11). The budding index of these reconstructed strains are comparable to those of both wild-type and Rec8-expressing strains (S12 Fig), confirming that none of the mutations affect the timing of Start after release from a G1 arrest.

We asked how Rec8 altered genome replication and how individual adaptive mutations restored the tempo of genome replication to that of wild-type cells. We constructed a whole genome replication profile by genome sequencing multiple time points of a synchronized yeast population proceeding through S phase [50]. By analyzing changes in read depth during S phase, we calculated $T_{rep}$, the time at which 50% of cells in a population complete replication at a given genomic locus. The profile of $T_{rep}$ across the yeast genome reveals the dynamics of replication: the peaks mark points at which replication initiates, namely a fired replication origin, and the slopes show the speed of the replication forks that move away from the origins. We compared the replication profiles of wild-type, *scc1Δ*, Rec8-expressing, and reconstructed strains that express Rec8 and carry a single adaptive mutation in one of three genes: *ssn3Δ*, *esp1-P15*, or *smc1-P15*. Compared to wild type, the Rec8-expressing strain fired many but not all replication origins earlier, and on average an origin fired four minutes earlier in the Rec8-expressing cells. In contrast, the temporal order of origin firing and the speed of replication forks were similar to those in wild type. The replication profile of *scc1Δ* strain was

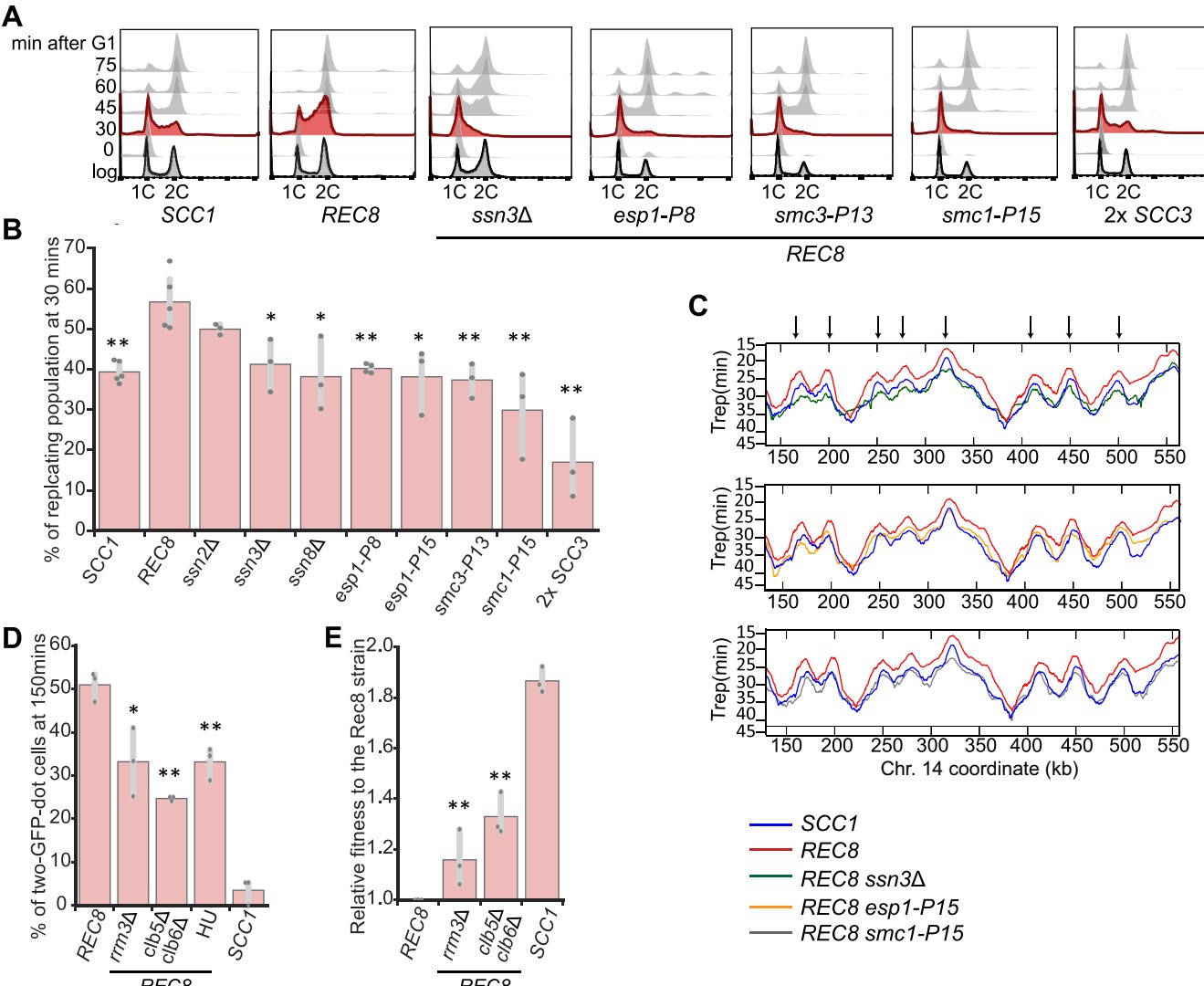

**Fig 6. Delaying genome replication partially improves sister chromosome cohesion in Rec8-expressing cells. (A)** Cell cycle profiles of wild-type, the Rec8-expressing strain, and the Rec8-expressing strains carrying a single reconstructed mutation. Cells were released from a G1 arrest as described in Fig 1D. Flow cytometry profiles are shown at the indicated times and profiles at 30 minutes are labeled in red. **(B)** Quantitation of the fraction of replicating cells in strains carrying a single reconstructed mutation at 30 minutes after release from a G1 arrest. The replicating subpopulation was measured as the fraction of the population between the G1 peak and the G2/M peak. The darker gray points represent the values of three biological replicates, and the thinner gray bar represents one standard deviation on each side of the mean of these measurements. The statistical significance between data from the Rec8-expressing strain and each mutation-reconstructed strain was calculated by two-tailed Student $t$ test, $^{*}p < 0.05$, $^{**}p < 0.01$. **(C)** The replication profiles of wild-type, the Rec8-expressing strain, and the Rec8-expressing strains with single reconstructed mutations (ssn3Δ, esp1-P15, or smc1-P15). Replication dynamics is expressed as $T_{rep}$ (shown on the y-axis), the time at which 50% of cells in a population complete replication at a given genomic locus. The mean replication profile of two experiments on one part of Chromosome 14 is shown. The replication profile of each strain is color-coded. An arrowhead represents a fired replication origin. We confirmed that the different strains exited from G1 at the same time by monitoring their budding index over time (S15 Fig). **(D)** Genetically and chemically perturbing genome replication improves sister chromosome cohesion. The Rec8-expressing rrm3Δ and clb5Δ clb6Δ strains were assayed for sister chromosome cohesion as described in Fig 1C. Hydroxyurea was added in YPD as Rec8-expressing cells entered the cell cycle. The percentages of cells with two GFP dots in mitotically arrested populations (150 minutes after G1) are shown. The darker gray points represent the values of three biological replicates and the thinner gray bar represents one standard deviation on each side of the mean of these measurements. The statistical significance between data from the Rec8-expressing strain and each mutant strain was calculated by two-tailed Student $t$ test, $^{*}p < 0.05$, $^{**}p < 0.01$. **(E)** The effect of rrm3Δ and clb5Δ clb6Δ on the fitness of the Rec8-expressing strain. The darker gray points represent the values of three biological replicates, and the thinner gray bar represents one standard deviation on each side of the mean of these measurements. The statistical significance between data from the Rec8-expressing strain and each mutant strain was calculated by two-tailed Student $t$ test, $^{**}p < 0.01$. Data associated with Fig 6B, 6D, and 6E can be found in S1 Data. clb, cyclin B 5; esp1, extra spindle pole bodies 1; GFP, green fluorescent protein; HU, hydroxyurea; Rec8, recombination 8; rrm3, rDNA recombination mutation 3; SCC, sister chromosome cohesion; smc1, structural maintenance complex 1; ssn3, suppressor of snf1; YPD, yeast extract, peptone, and dextrose.

indistinguishable from that of wild type (S13 Fig). Three adaptive mutations we examined delayed origin firing to various degrees: *ssn3Δ* or *esp1-P15* almost restored the pattern of origin firing to that of wild type. *smc1-P15* made the firing of many origins later than those of wild type (Figs 6C and S14). Overall, the replication profiles of these reconstructed strains are more similar to the genome-wide pattern of wild type rather than that of the Rec8-expressing strain. We concluded that expressing Rec8 in mitosis advances the timing of origin firing, and therefore Rec8-expressing cells begin and finish genome replication earlier than wild type. Mutations in genes encoding the Cdk8 complex, cohesin, and separase restore the S-phase onset of Rec8-expressing strains by delaying origin firing. In Scc1-expressing cells, deleting the genes encoding the subunits of the Cdk8 complex altered the progression of S phase and led to an 8%–11% fitness reduction (S16 Fig), demonstrating that transcriptional mediator affects genome replication, even in the absence of Rec8, through an uncharacterized mechanism.

The correlation between the restored timing of genome replication and improved sister chromosome cohesion suggests that the dynamics of genome replication affect cohesion. Cohesin must be loaded onto chromosomes prior to or concomitant with the passage of the replication fork to be converted into functional cohesin, and delaying origin firing promotes the establishment of cohesive linkages near centromeres in a kinetochore mutant defective in cohesin accumulation at centromeres [51]. Based on these observations, we hypothesized that slowing genome replication would improve Rec8-dependent sister chromosome cohesion. To test this idea, we asked if manipulations that slow genome replication improve sister chromosome cohesion and the fitness of the Rec8-expressing strain, potentially by allowing more time for cohesin to load prior to replication fork passage. We found that both decreasing origin firing and slowing the movement of replication forks improved sister cohesion. Removing the two S phase cyclins, cyclin B (Clb)5 and Clb6, which delays replication origin firing [52,53], or reducing the speed of replication forks by removing Rrm3, a helicase involved in DNA replication [54], halved the sister cohesion defect in Rec8-expressing cells (Fig 6D) and increased their fitness by 31% (*clb5Δ clb6Δ*) and 24% (*rrm3Δ*) (Fig 6E). Slowing genome replication with hydroxyurea (HU), which lowers the concentration of deoxyribonucleotide triphosphates (dNTPs), also improved sister cohesion and fitness (Figs 6C and S17). We infer that in Rec8-expressing cells, the Cdk8 and cohesin-related mutations exert at least part of their effects by delaying genome replication and thus improving sister cohesion (Fig 7).

## Discussion

We used experimental evolution to study how cells adapt to the demand that a protein performs in a different biological function. Budding yeast adapt to use the meiotic kleisin, Rec8, which normally functions in meiosis, to maintain the sister chromosome linkage required for accurate mitotic chromosome segregation. Whole genome sequencing of the adapted populations identified a mutation in *REC8* in one population and repeated, adaptive mutations in three functional modules: the transcriptional mediator complex, cohesin structure and regulation, and cell cycle regulation. Individually, these mutations delay the timing of genome replication, improve sister cohesion, and increase the fitness of the ancestral, Rec8-expressing strain. Engineering mutations that delay the firing of replication origins or slow the speed of replication forks into the ancestral, Rec8-expressing strain increased sister chromosome cohesion and fitness, demonstrating a causal link between genome replication and sister chromosome cohesion. Our work suggests that mutations, both in the components and regulators of cohesin and in other proteins, which were not previously implicated in chromosome cohesion, improve the ability of the meiotic kleisin to function in mitosis, despite the passage of a billion years since the divergence between mitotic and meiotic kleisins.

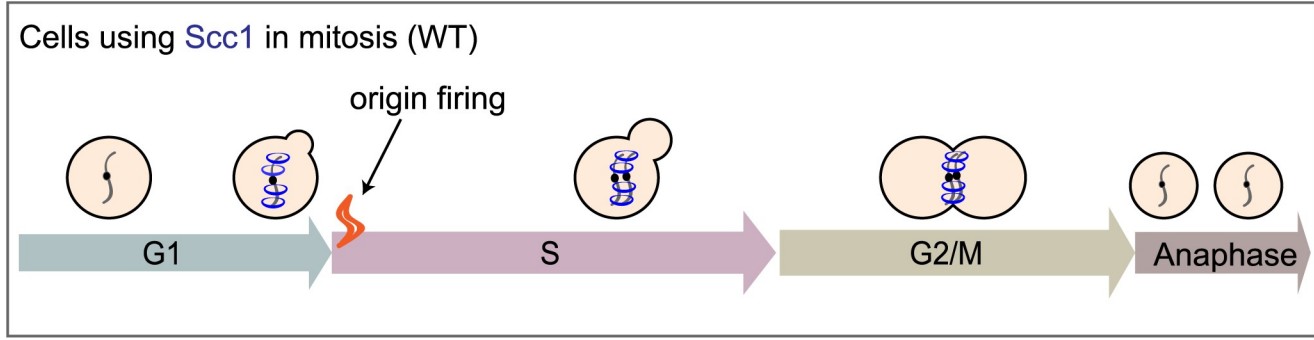

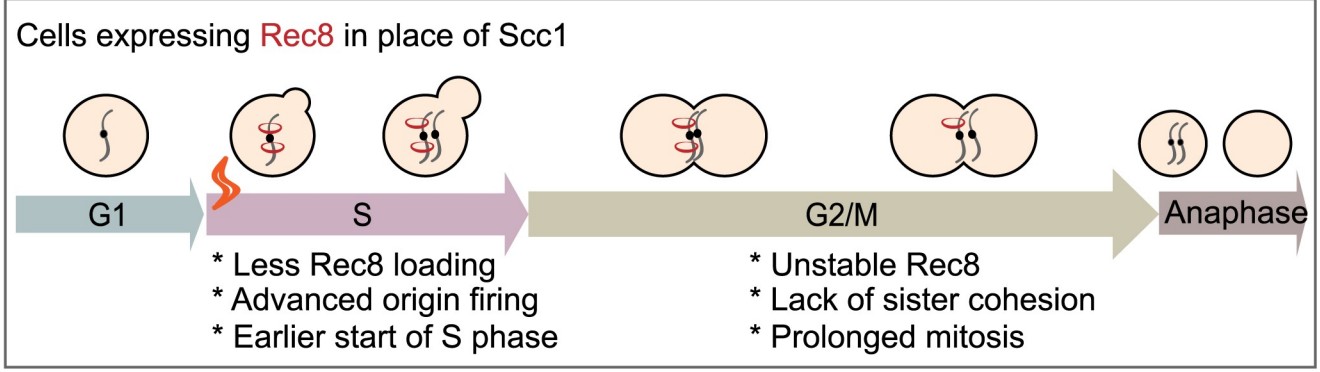

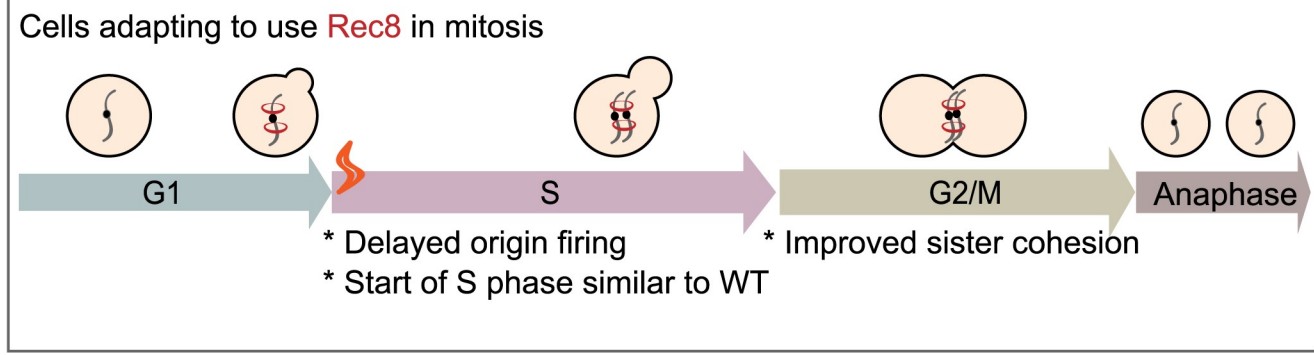

**Fig 7. Summary of the mechanism that allows budding yeast to use the meiotic kleisin, Rec8, for mitosis.** Yeast cells expressing Rec8 in place of Scc1 cannot build robust cohesion to hold sister chromosomes together before anaphase due to the weak association of cohesin with chromosomes and Rec8 protein instability. Rec8-expressing cells induce earlier firing of replication origins compared to what wild type does and start S phase earlier. After experimental evolution, adaptive mutations in different functional modules delay origin firing and improve sister chromosome cohesion of Rec8-expressing cells, potentially by allowing more time for Rec8-containing cohesin to load onto chromosomes prior to the passage of replication forks. Rec8, recombination 8; Scc1, sister chromosome cohesion 1; WT, wild type.

What allows Scc1 and Rec8 to participate in two different forms of chromosome segregation? Cells that are forced to use Rec8 in mitosis have multiple defects that account for their reduced fitness. In G1, ectopically expressed Rec8 associates more weakly with chromosomes than Scc1 does. This defect may reduce the ability to productively load Rec8-containing cohesin on chromosomes. In mitotically arrested cells, Rec8 is less stable than Scc1, binds less well to peri-centromeres and chromosomal arms, and shows increased binding specifically at core centromeres. We suggest that the reduced pericentromeric binding in mitotically arrested cells destabilizes sister chromosome cohesion. Cohesin binding to chromosomes is initiated by

cohesin loading, which is either specifically targeted to centromeres and dependent on the Ctf19 kinetochore complex [55,56], or generally loaded genome-wide. Following loading, cohesin translocates to other parts of chromosomes [57,58]. In mitosis, Rec8's increased binding at core centromeres and reduced binding at peri-centromeres suggest that Rec8-containing cohesin can be targeted to centromeres but cannot translocate efficiently to the peri-centromeric borders, where most of Scc1-containing cohesin accumulates to generate linkages between sister chromosome [59]. Rec8's genome-wide binding pattern suggests Rec8 does not associate with chromosomes in the same way as Scc1. The reduced stability of Rec8 in mitosis is partially due to separase activity: populations acquired hypomorphic alleles of separase, and a temperature-sensitive separase mutant stabilized Rec8 in mitotically arrested cells. In meiosis, the difference in the stability of the linkage with the two forms of cohesin is reversed: Scc1 near centromeres is not protected from separase activity, whereas Rec8 is [25]. Because the other cohesin subunits and cohesin regulators are present in both the mitotic and meiotic cell cycles, these differences must be due to differential modification of the known cohesin regulators or additional components that interact differently with Rec8 and Scc1. Why is Rec8 unable to fully substitute for Scc1 in the mitotic cell cycle? Our results do not distinguish between two possible trajectories that led to the functional separation between the current two kleisin paralogs: (i) A single kleisin cannot easily support the different forms of chromosome segregation seen in mitosis and meiosis, and this incompatibility forced the two kleisin paralogs to diverge from each other, and (ii) mutations that impaired Rec8's ability to support mitosis accumulated by genetic drift rather than selection. Our work reveals the power of a variety of adaptive mutations affecting diverse modules to alter the ability of a protein to participate in a biological function. How specific are the benefits of the mutations that we observed to the cells using the meiotic kleisin in mitotic cell cycle? This question could be answered by investigating their effects in two contexts in wild-type cells: in mitosis that used the mitotic kleisin, Scc1, and in meiosis, with Rec8 expressed from its normal promoter.

What accounts for the genes that acquired adaptive mutations and the order in which mutations appear? We argue that the answer is a combination of the benefit conferred by mutations in a gene and the target size for these beneficial mutations. The mutations in the transcriptional mediator complex and genes regulating the G1-to-S transition are likely to be strong loss-of-function mutations: Many of the mutations are nonsense mutations and gene deletions mimic the effect of the evolved mutations. Two arguments suggest that the mutations in cohesin and its regulators are different: These are essential genes and their mutations accumulate later in evolution than the mediator mutations even though they produce similar fitness increases. This delay is consistent with the target for adaptive, cohesin-related mutations being smaller than the target for inactivating mediator. Genetic evidence suggests that the mutations in separase are mild loss-of-function mutations, but the effect of mutations in Smc1 and Smc3 are unclear. Mutations in these proteins can directly alter their interactions with kleisin, but any mutation that disrupts the essential biochemical activity of cohesin will be lethal. We argue that the number of mutations that change the regulation of the cohesin complex but not its essential activity is small, explaining the later accumulation of these mutations. We see frequent aneuploidy for Chromosome 9 and two different examples of segmental aneuploidy, each occurring in only one population. The prevalence of aneuploidy compared to segmental duplication in our evolution experiment is likely to be explained by frequent chromosome mis-segregation in Rec8-expressing cells. We demonstrated that gaining an extra copy of Chromosome 9, increasing the copy number of *SCC3*, could be beneficial to Rec8-expressing cells; however, at generation 1,750, five out of twelve populations lost one extra copy of Chromosome 9, suggesting the benefit of Chromosome 9 disome is transient in some evolved populations. We did not find any mutation or increased copy number of *SCC3* in

these five populations, suggesting that the loss of Chromosome 9 disome is not associated with the alteration of *SCC3*. We speculate that the fitness effect of gaining an extra copy of *SCC3* competes with the cost of unbalanced gene caused by an extra copy of Chromosome 9 and the balance between the cost and benefit changes during the evolutionary trajectories of our populations. Because the segmental duplications were rare, we cannot conclude that they are beneficial. The segmental duplication on Chromosome 5 could promote adaptation of Rec8-expressing cells because the duplicated region contains *SCC4*, one component of the cohesin loading complex. Duplication of this gene improves sister chromosome cohesion in yeast cells adapted to replication stress [36].

We argue that considering the target size for different mutations explains why we saw only one mutation in Rec8 at generation 1,750. Because Rec8 and Scc1 have diverged substantially roughly a billion years, it may require multiple, simultaneous amino acid substitutions in Rec8 to improve its ability to hold mitotic sister chromosomes together. Even if single amino acid substitutions in Rec8 can improve its function in mitosis, there are unlikely to be many such mutations, and the selective advantage conferred by individual mutations is likely to be modest. In our experiment, mutation in Rec8 only occurred once, so we don't know if it was selected. In contrast, the target size for inactivating mutations, such as those in mediator and G1-to-S regulators, are large. If the mutations in *SMC1*, *SMC3*, and *ESP1* reduce some aspect of their function, the target size for mutations in these genes will be larger than the target size for mutations that improve Rec8's mitotic function. Our results are consistent with other studies in which loss-of-function mutations are the first step in adaptation in laboratory evolution experiments [40, 60–62].

Mutational target size is likely to explain why adaptive mutations often occur outside the gene whose product is being asked to perform a different function. When *Escherichia coli* is experimentally evolved to use an enzyme that normally participates in proline synthesis, proline mutant A (ProA), to catalyze a similar reaction in arginine synthesis, most of the adaptive mutations are in other genes of arginine synthesis pathway, not in ProA [63]. Mutations outside the focal gene are also found when proteins are asked to perform the same function in a novel cellular environment. Thus, *E. coli* adapts to use orthologs of the *folA* gene, which encodes dihydrofolate reductase, via mutations in genes responsible for protein degradation rather than mutations in the *folA* ortholog [64]. We suggest that evolutionary changes in a protein's function reflect a mixture of changes in its sequence and expression, changes in the proteins that it physically interacts with, and changes in other proteins that contribute to the biological function under selection. The number and diversity of these connections makes it difficult to predict the evolutionary trajectories that populations will follow as proteins are selected to perform new functions. Thus, evolutionary repair experiments are a strategy to learn more about the factors that regulate protein function and reveal previously unknown links between different functional modules.

Mutations in three functional modules, transcriptional mediator, chromosome cohesion, and cell cycle regulation, improve the fitness and chromosome segregation of Rec8-expressing cells. We used double mutants to probe the interactions between these modules, scoring both fitness and sister cohesion. Most pairs of mutations interacted roughly additively for both phenotypes, with some exceptions: Double mutations with *esp1-P15* were substantially fitter than the additive expectation and double mutations with *mbp1Δ* increased fitness but did not improve sister cohesion, suggesting that this mutation has effects on both sister cohesion and some other function. None of the adaptive mutations increase the total level of Rec8 in mitotically arrested cells, suggesting that they likely alter the ability of Rec8-containing cohesion to form and maintain the linkages that hold sister chromosomes together.

Our work reveals a new regulatory link between sister cohesion and genome replication. The Rec8-expressing strain advances the timing of genome-wide origin firing and completes

genome replication earlier than wild type. All the adaptive mutations we tested delay origin firing and improve sister cohesion. We found deletion of genes in the Cdk8 complex, separase mutation, and cohesin mutation all restore the pattern of origin firing towards that of wild type. This result is consistent with multiple populations acquiring mutations that inactivate genes (*MBP1*, *CLN2*, *SWI4*, and *SWI6*) that promote the passage from G1 to S phase. We speculate that slowing genome replication can promote sister chromosome cohesion in the Rec8-expressing strain. We tested the causality of this linkage using mutants that reduce origin firing or slow replication forks: Both manipulations improve the fitness and sister chromosome cohesion of Rec8-expressing cells, demonstrating that slower replication raises fitness. A recent study demonstrated that Mad2, the spindle checkpoint protein, regulates S phase by promoting translation of Clb5 and Clb6 [65], potentially explaining why *mad2Δ* also restores the S phase of Rec8-expressing cells to that of wild type. Our work demonstrates that mitotic sister chromosome cohesion can be improved by mutations that delay genome replication. The simplest explanation of this effect is that delaying the timing of replication allows more time for Rec8-containing cohesin to associate with chromosomes either before or during the passage of the replication fork.

Our work leads to a number of questions. Why does expressing Rec8 advance the timing of origin firing, while removing Scc1 has no effect? How do mutations in transcriptional mediator and cohesin-related genes affect replication? Are the effects of Rec8 on replication in the mitotic cycle related to its reported ability to stimulate genome replication in the meiotic cycle [66]? One possibility is that Rec8 affects replication by altering the level of chromosome-bound Rad53, a checkpoint protein that can regulate origin firing [67] and whose role has been reported to be altered in meiosis [68]. In early S phase, replication initiation is orchestrated by a series of molecular interactions: the proteins that activate DNA replication, like S-phase CDK and Dbf4-dependent kinase (DDK), recruit several initiation factors to form an activated helicase complex [69]. Cells can control the timing of origin firing by modifying the activity of these activators, limiting the dosage of replication initiation factors [70] or changing local chromatin structure, which affects how easily these regulators can access a given origin [71,72]. Further research is needed to determine whether Rec8 affects replication by increasing the expression of genes that control replication initiation or altering chromatin structure to make replication origins more accessible to initiation factors, or some combination of both. The combination of overexpressing four initiation factors and reducing chromatin compactness accelerates the firing of late replication origins, demonstrating that these factors can alter the dynamics of replication [70].

The most pressing question is how mutations in the transcriptional mediator complex, the major target of early adaptive mutations, alter the timing of genome replication and increase the fitness of Rec8-expressing cells. There are suggestions that transcriptional mediator is involved in genome replication. In budding yeast, genes of the Cdk8 complex have genetic interactions with genes that trigger replication initiation (*DBF4*, *DPB11*, *SLD3*, and *CDC7*) and core helicase components (*SLD5*) [73]. In fission yeast [74] and mammalian cells [75], mutations of the Cdk8 complex are reported to alter genome replication, suggesting our finding that the Cdk8 module is involved in genome replication is not species specific, although the detailed mechanism remains unknown.

Overall, this evolution experiment shows that mutations outside the meiotic kleisin, Rec8, improve its ability to support mitotic chromosome segregation. We speculate that the ability of mitotic and meiotic kleisins to participate in the related, but different, processes of mitotic and meiotic chromosome segregation evolved through a mixture of changes in kleisin itself and changes in other functional modules that regulate sister chromosome cohesion directly or indirectly. At least in laboratory experiments, the size and complexity of this molecular

network provide a much larger target for mutations that alter the biological function of kleisin than the target presented by kleisin itself. More generally, we suggest that the ability of proteins to participate in novel functions, such as the divergence in the roles of paralogous proteins, depends on a mixture of mutations in the proteins themselves and the proteins that they directly or indirectly interact with.

## Materials and methods

### Yeast strains, plasmids, and growth conditions

All yeast strains are derivatives of W303, and their genotypes are listed in S2 File. For the yeast strain with a GFP-labeled Chromosome 5 and P$_{GAL1}$-SCC1, yPH344 and yPH345 are haploid strains derived from a diploid strain made from a cross between FY1456 (the same strain as K7289, a gift from Dana Branzei) and yPH36. yPH346 is a haploid strain derived from a diploid strain made from a cross between FY1456 and yPH115. Strains carrying *REC8* integrated at the endogenous *SCC1* locus were generated by homologous recombination: A *REC8*-3xHA fragment was amplified from the plasmid pFA6a-*REC8*-3xHA-*KANMX4*, fused with 500-bp upstream and downstream DNA fragments of SCC1 coding sequence by PCR, and recombined with the *SCC1* genomic locus. Strains used in the evolution experiment were generated from yPH280 by plasmid shuffling [76]. Standard rich media, YPD (1% yeast extract, 2% peptone, and 2% D-Glucose) was used for the evolution experiment. Growth conditions for each experiment are specified in the figure legends. Raffinose and galactose were used at 2%. Benomyl was used at 30 μg/mL. Cycloheximide was used at 35 μg/mL. HU was used at 12.5 mM. α-Factor was used at 10 μg/mL for *bar1* strains and at 100 μg/mL for *BAR1* strains. Methionine was used at 8 mM.

### Experimental evolution

The haploid strain used in the evolution experiment was *MAT*α *scc1*Δ pRS*414*-P$_{SCC1}$-*REC8*-HA. To force yeast cells to depend on Rec8 for mitotic growth, five clones of yPH280 (*MAT*α *scc1*Δ pRS*414*-P$_{SCC1}$-*REC8*-HA pRS*416*-*SCC1*) were cultured in YPD to lose pRS416-*SCC1*, and cells without the *SCC1*-bearing plasmid were selected by the growth on 5-FOA plates. For each of the five clones, three independent 5-FOA resistant colonies were chosen, giving rise to the 15 ancestral clones in the evolution experiment. Each ancestral clone was cultured in 3 mL YPD to reach $10^8$ cells/mL at 30°C and diluted 1:6,000 into a tube with 3 mL fresh YPD and incubated for 48 hours (before generation 375) or 24 hours (after generation 375). Each subsequent cycle used the same dilution. We estimated an effective population size of $6.3 \times 10^5$ cells using the formula [77], $N_e = N_o \times g$, in which $N_o$ is the initial population size ($5 \times 10^4$ cells), and $g$ is the number of generations, 12.6, during one cycle. After every 10 cycles, 1 mL culture was mixed with 500 μL 80% glycerol and frozen at −80°C. The evolution experiment was continued for 1,750 generations.

### Fitness measurement by competition assay

An ancestral P$_{SCC1}$-*REC8* strain that expressed a fluorescent protein, mCitrine, under the *ACT1* promoter was used as the reference strain (yPH447) in fitness competition assays with evolved populations and reconstructed strains carrying a single evolved mutation. For scoring fitness and sister cohesion in the same strain carrying a GFP-labeled Chromosome 5, a Rec8-expressing strain with P$_{ACT1}$-*mChrerry* (yPH472) was used as the reference strain.

All sample strains and the reference strain were grown to <$10^7$ cells/mL in YPD; cell density was measured using a Coulter Counter (Beckman Coulter). At the first time point, samples

strains, either evolved strains or reconstructed strains carrying evolved mutations, were mixed with the reference strain at a ratio of 1:10. The initial cell mixture was diluted to $5 \times 10^4$ cells/mL in YPD and grown for 24 hours. At the second time point, the cell density was usually around $5 \times 10^6$ to $10^7$ cells/mL. Cultures were diluted to $5 \times 10^4$ cells/mL to grow another 20–24 hours as the third time point. At each timepoint, $5 \times 10^4$ cells of each mixed culture were transferred to single wells of a 96-well plate with U-shaped bottom for flow cytometry. A BD LSRFortessa FACS machine equipped with High Throughput Sampler was used to collect 30,000 cells to quantify the ratio of the sample and the reference strain. The FACS data were analyzed using the FlowJo10.4.1 software. In addition to being mixed with sample strains, the reference strain was cultured separately to estimate the number of generations in an experiment. Each experiment was conducted in technical triplicates, and the fitness of each sample strain was measured in at least three independent experiments.

To calculate relative fitness, *w*, of each sample strain to the reference strain, we followed this formula: $w = 1+s$,

where *s* is selection coefficient: $s = \dfrac{\ln\left(\frac{Sample}{Referece}\right)_g - \ln\left(\frac{Sample}{Reference}\right)_0}{g}$,

in which *g* is the number of generations and $\frac{Sample}{Reference}$ is the ratio between a sample strain and a reference strain [78].

## ChIP and qPCR

We followed the protocol of calibrated ChIP [79] to precipitate chromosome-bound kleisins. First, cell quantities were measured by multiplying culture volumes by optical density at 600 nm ($OD_{600}$) and this product is referred as O.D. units. A total of 20 O.D. of *S. cerevisiae* cells was cross-linked with 1% formaldehyde for 30 minutes at 25°C. Each *S. cerevisiae* cell pellet was mixed with 15 O.D. of the cross-linked *S. pombe* cells that expressed an epitope-tagged version of the Scc1 homolog (*RAD21*-HA). The inclusion of the fission yeast cells served as control to normalize technical variations between samples. This mixture was resuspended in ChIP lysis buffer A (50 mM HEPES-KOH at pH 7.5, 0.1 M NaCl, 1 mM EDTA, 150 mM NaCl, 1% TritonX-100, 0.1% Sodium Deoxycholate, and 1x protease inhibitor [Roche]) and further lysed by bead beating (BioSpec Products) with 0.5-mm glass beads (Biospec Products). To shear chromatin, a Covaris S220 instrument was used with the following program: peak incident power, 175; duty factor, 10%; cycle per burst, 200; treatment time, 250 seconds. After shearing, the cell lysate was centrifuged at 16,000*g* at 4°C for 20 minutes to collect the supernatant containing protein-bound, sheared chromatin. To pull down the fraction of chromatin bound by kleisin, 15 μL prewashed dynabeads, ProteinG (Invitrogen), and 7.5 μL anti-HA antibody (12CA5, Invitrogen) was added in 1 mL lysate and incubated at 4°C overnight. After immunoprecipitation, the beads were washed with ChIP wash buffer I (50 mM HEPES-KOH at pH 7.5, 0.1 M NaCl, 1 mM EDTA, 274 mM NaCl, 1% TritonX-100, 0.1% Sodium Deoxycholate), ChIP wash buffer II (50 mM HEPES-KOH at pH 7.5, 0.1 M NaCl, 1 mM EDTA, 500 mM NaCl, 1% TritonX-100, 0.1% Sodium Deoxycholate), ChIP wash buffer III (10 mM Tris/HCl pH 8.0, 0.25 M LiCl, 1 mM EDTA, 0.5% NP40, 0.5% Sodium Deoxycholate), and TE (10 mM Tris/HCl pH 8.0, 1 mM EDTA).

To process the sample for quantitative PCR, immunoprecipitated chromatin and a 1:100 dilution of the input chromatin were separately recovered by boiling with a 10% Chelex-100 resin (BioRad) before treating with 25 μg/mL Proteinase K at 55°C for 30 minutes. Samples were boiled again to inactivate Proteinase K, centrifuged, and the supernatant was subjected to qPCR on ABI 7900 using PerfeCTa SYBR Green FastMix ROX (Quanta BioSciences). The sequences of primers used for qPCR are listed in S3 Table. To calculate the enrichment of pulldown DNA in total input chromatin, we used the following formula: $\frac{ChIP}{input} = E^{-\Delta Ct}$,

$\Delta Ct = Ct_{ChIP} - (Ct_{input} - \log_E(dilution\ factor))$, in which $E$ is primer efficiency and *dilution factor* is 100. Enriched percentage of input is $100 \times \frac{ChIP}{input}$. At each cohesin binding site, the final enriched percentage of input is calibrated to the enriched percentage of input at a peri-centromeric site of the *S. pombe* genome.

For ChIP-Seq, chromatin was immunoprecipitated as described above, and purified chromatin was subjected to DNA end repair and dA tailing to make the sequencing library as described [79]. Samples were sequenced on a MiniSeq with 75 base paired–end reads (Illumina, San Diego, CA). ChIP-Seq data sets are deposited with the NCBI Gene Expression Omnibus under the accession number GSE 141598. Scripts and workflows used to analyze ChIP-Seq data are stored on the github repository (github.com/PhoebeHsieh-yuying). The enrichment of kleisin binding on chromosomes is visualized by Integrated Genome Viewer [33].

## Cell cycle progression by flow cytometry

Yeast strains with $P_{GAL1}$-*SCC1* were grown in YEP containing 2% galactose to log phase. To synchronize populations in G1, cells were washed and diluted in YEP containing 2% raffinose and 100 μg/mL α-factor for 2 hours. G1 synchronization was confirmed by checking that the percentage of cells that had formed mating projections (shmoos) was over 90% using a light microscope (MICROPHOT-SA, Nikon). To restart the cell cycle without *SCC1* transcription, cells were washed with YEP containing 50 μg/mL pronase (Sigma-Aldrich) twice and resuspended in YPD containing 50 μg/mL pronase at 30˚C. A total of 1 mL of cells was collected and fixed by 70% ethanol at G1 and several time points after growth in YPD according to the figure legend of each experiment. Subsequently, fixed samples were treated with 0.4 mg/mL RNase A (Sigma-Aldrich) at 37˚C overnight, followed by 1 mg/mL proteinase K (Sigma-Aldrich) treatment at 50˚C for 1 hour. DNA was stained with 1 μM Sytox Green solution (Invitrogen). Prior to flow cytometry, the stained cells were sonicated for 30 seconds with 70% intensity by BRANSON Ultrasonics Sonifier S-250. A total of 10,000 cells were collected using a BD LSRFortessa FACS machine (Becton Dickinson). The FACS data were analyzed using FlowJo10.4.1. To quantify the fraction of replicating cells in the population at 30 minutes after release from the G1 block, the Watson (Pragmatic) model built in the cell cycle analysis tools of FlowJo was used to calculate the portion of cells that had completed replication (G2 peak), were undergoing genome replication (S phase), and were still in G1.

## Sister chromosome cohesion assay and microscopy

Strains with $tetO_{112}$ array integrated at the *URA3* locus and *tetR-GFP* were used for assaying cohesion between sister chromosomes [29]. To examine sister chromosome cohesion in one cell cycle, sample preparation followed the same procedure as the cell cycle progression experiment except that cells were released from their G1 arrest into YPD containing 30 μg/mL benomyl to arrest them once they reached mitosis. At each time point, cells were fixed in 4% paraformaldehyde for 15 minutes, washed, and stored in a storage solution (1.2 M sorbitol, 0.1 M $KH_2PO_4$/$K_2HPO_4$) at 4˚C. Images were taken with a 100× objective on a Nikon inverted Ti-E microscope with a Yokagawa spinning disc unit and an EM-CCD camera (Hamamatsu ImagEM); GFP was excited with a 488-nm laser with 25% laser power. For each image, a z-stack was taken with 41 z-steps spaced 0.5 μm apart. Images were analyzed by the Fiji distribution of ImageJ [80]. For each experiment, at least 100 cells are analyzed. Three independent experiments were conducted for each strain.

## Measurement of protein abundance and protein stability

The protein-containing extracts from cell pellets were prepared by alkaline lysis [81] and analyzed on western blots. Protein extracts were resuspended with SDS sample buffer (10 mM Tris pH 6.8, 2% SDS, 10% glycerol, 0.004% bromophenol blue, and 2% β-mercaptoethanol) and boiled at 100˚C for 5 minutes. Rec8 and Scc1 were all tagged with 3xHA at the C terminus of coding sequences, and anti-HA antibody (3F10, Roche) was used to detect the abundance of kleisin proteins. The abundance of hexokinase (Hxk1) was monitored as a loading control using an anti-Hxk1 antibody (USBiological Life Sciences, H2035-01). Super-Signal West Dura reagent (Thermo Scientific) was used for developing chemiluminescent signals. Chemiluminescent signals were detected by Azure Sapphire Biomolecular Imager, and quantification of protein abundance was analyzed by ImageStudioLite (www.licor.com/bio/image-studio-lite/).

To measure protein stability in mitosis, cells were arrested at metaphase in YPD containing 30 μg/mL benomyl for 2 hours and treated with cycloheximide at 35 μg/mL to inhibit protein synthesis. One milliliter of cells was collected for to prepare samples for western blotting at the indicated time points after adding cycloheximide. Protein half-life was analyzed as described [82].

## Whole genome sequencing and analysis

Genomic DNA was prepared as described [61]. DNA sequencing libraries were prepared using the Illumina Nextera DNA library Prep kit as described [83]. The sequencing was done on an Illumina HiSeq 2500 with 125 base paired–end reads or Illumina NovaSeq with 150 base paired–end reads. Whole genome sequencing data were processed as described [61]. The Burrow-Wheeler Aligner (bio-bwa.sourceforge.net) was used to map DNA sequences to the *S. cerevisiae* reference genome r64, downloaded from *Saccharomyces* Genome Database (www.yeastgenome.org). The resulting SAM (Sequence Alignment/Map) file was converted to a BAM file, an indexed pileup format file, using the samtools software package (samtools.sourceforge.net). GATK (gatk.broadinstitute.org/hc/en-us) was used to realign local indels, and Varscan (varscan.sourceforge.net) was used to call variants. Mutations were identified using an in-house pipeline (github.com/koschwanez/mutantanalysis) written in Python. Variants that differ between the ancestral and evolved genome >10%, a threshold above average sequencing error, are called as mutations (S3 File), and any mutation present in >90% of sequencing reads in the evolved genome is defined as a fixed mutation. In our pipeline, mutations can be found in both coding and noncoding sequences. In this study, we focused on mutations that cause non-synonymous substitution in the coding sequence (S1 File). Scripts and workflows used to find evolved mutations are stored on the github repository (github.com/PhoebeHsieh-yuying). Whole genome sequencing data sets are deposited with the NCBI BioProject under the accession number PRJNA594153.

## Generation of reconstructed strains

Individual mutations from evolved strain were engineered into the targeted locus of the ancestral strain by homologous recombination. A DNA fragment containing targeted gene with the desired mutation, a selection maker (*HpHMX4* or *HIS3MX4*), and 300 bp downstream of the targeted gene was made by PCR. To measure the fitness effect of an evolved mutation in the ancestor, this DNA fragment was transformed into yPH280. The presence of the desired mutation was confirmed by Sanger sequencing. To measure the fitness effect of reconstructed mutations in the Rec8-expressing background, yeast cells were cultured in YPD to lose pRS416-*SCC1* and cells without *SCC1* plasmid were selected by the growth on 5-FOA plates.

To measure the effect of these reconstructed mutations on cell cycle progression and sister cohesion, the same transformation procedure was done in the $P_{GAL1}$-SCC1 $P_{SCC1}$-REC8 strain with GFP-labeled Chromosome 5 and phenotypes were assayed in the presence of glucose.

## Whole genome replication profile

Yeast cells were arrested in G1 with 2 μg/mL α-factor and low pH as described [84] and released for entering the cell cycle in the same procedure of flow cytometry experiments. A total of 1 mL of cells was collected separately for DNA content analysis and genomic DNA extraction at the following time points: 0, 10, 20, 30, 40, 50, and 60 minutes after G1 arrest. Whole genome sequencing libraries were made as previously described. Sequencing was done on an Illumina NovaSeq with 150 base paired–end reads. Two separate experiments were done for each strain. Whole genome sequencing data sets are deposited with the NCBI BioProject under the accession number PRJNA594153.

The analysis of genome replication profile was done as described [36,50]. Reads mapping and CNVs detection were processed as in [61]. We followed the script in [36] to analyze change in the CNVs at multiple time points during S phase to generate whole genome replication profile. First, the read depth of every 100-bp window is normalized to the median read depth of a sequenced genome to control for sequencing variation between samples. To allow intra-strain comparison at multiple time points, the normalized read depth was further scaled to the median of DNA content obtained by flow cytometry to generate relative coverage to the corresponding G1 genome. The resulting coverage was then averaged across multiple 100-bp windows and a polynomial data smoothing filter (Savitsky-Golay) was applied to the individual coverage profiles to filter out noise. Replication timing, $T_{rep}$, is defined as the time at which 50% of the cells in the population replicated a given region of the genome, which is equivalent to an overall relative coverage of 1.5×, since 1× corresponds to an unreplicated region and 2× to a fully replicated one. The replication timing $T_{rep}$ was calculated by using linear interpolation between the two time points with coverage lower and higher than 1.5× to compute the timing corresponding to 1.5× coverage. Final $T_{rep}$ were then plotted relative to their window genomic coordinates. Scripts and workflows used to generate whole genome replication profiles are stored on the github repository (github.com/marcofumasoni).

## Supporting information

**S1 Fig. Mitotic growth and chromosome segregation of the $P_{GAL1}$-SCC1 $P_{SCC1}$-REC8 strain.** We used a $P_{GAL1}$-SCC1 $P_{SCC1}$-REC8 strain to examine the effect of acutely expressing Rec8 as the sole kleisin. Cells were propagated in galactose-containing media, arrested in G1, and then released into glucose-containing media to repress Scc1. **(A)** The $P_{SCC1}$-REC8 strain grows poorly when *SCC1* expression is turned off. Left: Cells were grown in YEP containing 2% galactose to the same density and serially diluted on YEP containing 2% galactose or 2% glucose, in which the *GAL1* promoter was repressed by glucose. Right: The fitness of a $P_{GAL1}$-SCC1 $P_{SCC1}$-REC8 strain relative to that of wild type in YPD. The darker gray points represent the values of three biological replicates, and the thinner gray bar represents one standard deviation on each side of the mean of these measurements (two-tailed Student *t* test, ∗∗$p < 0.01$). **(B)** The fidelity of chromosome segregation of the Rec8-expressing strain is 30% lower than that of wild type. $P_{GAL1}$-SCC1 $P_{SCC1}$-REC8 cells were grown in YEP containing 2% galactose to log phase, transferred to YEP containing 2% raffinose and α-factor to repress *SCC1* expression and arrest them in G1, prior to release into YPD to resume cell cycle with *SCC1* expression repressed. Once cells had entered S phase, α-factor was added again to prevent cells entering a second cell cycle. Chromosome segregation fidelity was measured as the fraction of

G1-arrested cells in a population showing one GFP dot, representing one copy of Chromosome 5, after one mitotic cell division. At least 100 cells were imaged in each experiment. The darker gray points represent the values of two biological replicates, and the thinner gray bar represents one standard deviation on each side of the mean of these measurements. Data associated with S1A and S1B Fig can be found in S1 Data. *GAL1*, galactose metabolism 1; GFP, green fluorescent protein; Rec8, recombination 8; Scc1, sister chromosome cohesion 1; YEP, yeast extract and peptone; YPD, yeast extract, peptone, and dextrose.
(TIF)

**S2 Fig. Sister kinetochore biorientation is perturbed in $P_{SCC1}$-*REC8* cells.** The yeast strain $P_{MET}$-*CDC20-3xHA* $P_{GAL1}$-*SCC1-3xHA* *CEN15::LacO* $P_{CUP1}$-*GFP-LacI* *SPC42-mCherry* was transformed with a pRS415-based plasmid of $P_{SCC1}$- *SCC1*, $P_{SCC1}$-*REC8*, or an empty plasmid. Cells were cultured in CSM-Met-Leu containing galactose to log phase and switched to CSM-Met-Leu containing raffinose and α-factor to be synchronized in G1. Then, to repress the *SCC1* expression and arrest cells in metaphase, cells were released into YEP containing glucose and methionine for one cell cycle. The centromere of Chromosome 15 was marked by GFP and spindle pole bodies were labeled by *SPC42-mCherry*. Cells showing one or two GFP dots in the middle of two spindle pole bodies represent bi-oriented sister kinetochores under tension exerted by the spindle. The lack of sister chromosome cohesion leads to sister chromosomes of Chromosome 15 separating in prometaphase, resulting either in one GFP dot at each spindle pole body or two GFP dots at one of the spindle pole bodies. At least 100 cells were imaged and analyzed in each population. Illustrative microscopy images are shown in the right; the scale bar is 10 μm. Data associated with this figure can be found in S1 Data. CSM, complete synthetic media; GFP, green fluorescent protein; *SCC1*, sister chromosome cohesion 1; YEP, yeast extract and peptone.
(TIF)

**S3 Fig. The budding index of the Rec8-expressing strain is similar to that of wild-type and the *scc1Δ* strains.** Cells were arrested in G1 and then released as described in Fig 1C. The y-axis shows the fraction of budded cells in a population, measured as the budding index. At least 100 cells were examined at each time point for each experiment. The mean (solid line) and standard deviation (shaded region) of three biological replicates for each population are shown. Data associated with this figure can be found in S1 Data. Rec8, recombination 8
(TIF)

**S4 Fig. The genome-wide enrichment of Rec8 is lower than that of Scc1.** The ChIP-Seq data for all sixteen chromosomes (Chromosomes 3 and 15 are also shown in Fig 2D). Read depths calibrated to an internal control of the *S. pombe* genome are shown on the y-axis as reads per million (RPM, 0–300). The enrichment of Scc1 and Rec8 is shown in blue and red, respectively. The difference in the read depth between Scc1 and Rec8 is shown in the last track of each panel, in gray where Scc1's signal is higher than Rec8's, and in orange where Rec8's signal is higher than Scc1's. **(A)** ChIP-Seq data of individual chromosomes. **(B)** ChIP-Seq data of individual centromeres extending 20 kb on either side of the centromeres. Graphs were prepared using the Integrated Genomic Viewer [33]. ChIP-Seq, chromatin immunoprecipitation sequencing; Rec8, recombination 8; Scc1, sister chromosome cohesion 1.
(PDF)

**S5 Fig. Protein levels of Scc1 and Rec8 in cell extracts processed for ChIP experiments. (A)** Protein levels of two kleisins in mitosis. Cells were processed as described in Fig 2C and cell extracts were obtained by alkaline lysis prior to analysis by western blotting. Kleisin proteins were detected by anti-HA antibody and Hxk1 was used as a loading control. **(B)** Protein levels

of two ectopically expressed kleisins in G1. Cells were processed as described in Fig 2E and cell extracts were obtained by alkaline lysis prior to analysis by western blotting. The $P_{SCC1}$-SCC1-HA strain was used as a negative control because the endogenous *SCC1* gene is not expressed in G1. Hxk1 was used as a loading control. Raw images associated with S5A and S5B Fig can be found in S1 Raw Image. ChIP, chromatin immunoprecipitation; HA, hemagglutinin; Hxk1, hexokinase; Rec8, recombination 8; Scc1, sister chromosome cohesion 1.
(TIF)

**S6 Fig. Copy number data of five Rec8-expressing ancestors and two evolved populations, P4 and P7, that acquired segmental duplication. (A)** Chromosomal copy number of five Rec8-expressing ancestors. The copy number of each chromosome was calculated by normalizing the median read depth of each chromosome to the median read depth over the entire genome. Gray marks one copy, dark red marks two copies, and pink marks 1.25–1.75 copies, suggesting that part of the population was disomic. Data associated with this figure can be found in S1 Data. **(B)** The copy number data of Chromosome 4 of population P4 at generation 1,750. **(C)** The copy number data of Chromosome 5 of population P7 at generation 1,750. In **(B)** and **(C)**, copy numbers normalized to the median read depth of each sequenced genome are shown. Chromosomal regions showing copy number below 1.5 are marked with green, and regions showing copy number equal to 2 are marked with red. The position of centromeres is marked with a vertical gray line. The transposons on the edges of duplicated regions are annotated with horizontal black bars. Rec8, recombination 8.
(TIF)

**S7 Fig. *esp1-1* improves the growth of the Rec8-expressing strain at 30°C.** *esp1-1*, the known temperature-sensitive mutation that inactivates separase activity [32], was introduced to the Rec8-expressing strain by sporulating a heterozygous diploid strain ($P_{SCC1}$-SCC1/$P_{SCC1}$-REC8 ESP1/esp1-1). Haploid progeny carrying four different genotypes (*SCC1*, *SCC1 esp1-1*, *REC8*, and *REC8 esp1-1*) were selected. These four strains were subjected to serial dilutions and spotted on YPD. Their growth was measured at 25°C, 30°C, and 37°C. *esp1-1*, extra spindle pole bodies 1–1; Rec8, recombination 8; *SCC1*, sister chromosome cohesion 1; YPD, yeast extract, peptone, and dextrose.
(TIF)

**S8 Fig. Individual reconstructed mutations partially improve sister chromosome cohesion.** The time courses of sister chromosome separation for the experiment shown in Fig 5A, which presents the data at 150 minutes after release from a G1 arrest. At least 100 cells were imaged at each time point for each experiment. The mean and standard deviation of three biological replicates are shown in the solid line and shading, respectively. Data associated with this figure can be found in S1 Data.
(TIF)

**S9 Fig. Individual reconstructed mutations increase chromosome segregation fidelity of the Rec8-expressing strain.** Cells were prepared as in S1B Fig to examine the fidelity of chromosome segregation in a single mitotic cell division. Gene deletion for three components of the Cdk8 complex was used to approximate the effect of the mutations of these genes found in evolved populations. At least 100 cells were imaged in each experiment. The darker gray points represent the values of two biological replicates and the thinner gray bar represents one standard deviation on each side of the mean of these measurements. Data associated with this figure can be found in S1 Data. Cdk8, cyclin dependent kinase 8; Rec8, recombination 8.
(TIF)

**S10 Fig. Individual reconstructed mutations do not alter the Rec8 protein level in mitosis.**
Strains with individual reconstructed mutations were synchronized in G1, released into cell cycle, and then arrested in mitosis in YPD containing benomyl. Protein samples were collected by alkaline lysis and analyzed by western blotting. Both Scc1 and Rec8 were tagged with 3xHA at their C termini, and anti-HA antibody was used for their detection. Hxk1 was used as loading control. In the bar graph, the darker gray points represent the values of three biological replicates, and the thinner gray bar represents one standard deviation on each side of the mean of these measurements. Raw images and data associated with this figure can be found in S1 Raw Image and S1 Data, respectively. Hxk1, hexokinase; Rec8, recombination 8; Scc1, sister chromosome cohesion 1; YPD, yeast extract, peptone, and dextrose; 3xHA, 3 copies of hemagglutinin.
(TIF)

**S11 Fig. Individual reconstructed mutations partially restore the cycle progression profile of the Rec8-expressing strain towards that of wild type.** The full-time course data for the experiments summarized in Fig 6A, which shows the data from 0 to 75 minutes after release from a G1 arrest. Individual cohesin-related mutations, deletion of genes encoding the Cdk8 complex, and two integrated copies of *SCC3* were engineered separately into the $P_{SCC1}$-*REC8* $P_{GAL1}$-*SCC1* background. Cells were allowed to proceed through a synchronous cell cycle as in Fig 1D and were collected for fixation every 15 or 30 minutes following release from a G1 arrest to analyze their DNA content. Cdk8, cyclin dependent kinase 8; Rec8, recombination 8; *SCC3*, sister chromosome cohesion 3.
(TIF)

**S12 Fig. Individual reconstructed mutations do not delay the onset of cell cycle in Rec8-expressing cells.** Cells were arrested in G1 and then released as described in Fig 6A. The y-axis shows the fraction of budded cells in a population, measured as the budding index. Strains carrying single reconstructed mutations (*ssn2Δ*, *ssn3Δ*, *ssn8Δ*, *esp1-P8*, *esp1-P15*, *smc3-P13*, *smc1-P15*, or two copies of *SCC3*) are compared with the wild type and the Rec8-expressing strain. The mean (solid line) and standard deviation (shaded region) of three biological replicates for each strain are shown. Data associated with this figure can be found in S1 Data. Rec8, recombination 8; *SCC3*, sister chromosome cohesion 3.
(TIF)

**S13 Fig. The whole genome replication profiles of wild type, the Rec8-expressing strain, and the *scc1Δ* strain.** The mean replication profile of two experiments is shown. The replication profile of each strain is color-coded (*SCC1* in blue, *REC8* in red, and *scc1Δ* in green) and arranged by order of chromosome. The y-axis represents $T_{rep}$, the time at which 50% of cells in a population completes replication at a given genomic locus (See Materials and methods for detailed analysis). Rec8, recombination 8; *SCC1*, sister chromosome cohesion 1.
(TIF)

**S14 Fig. The whole genome replication profiles of the Rec8-expressing strain and reconstructed strains carrying a single evolved mutation, *ssn3Δ*, *esp1-P15*, or *smc1-P15*.** These replication profiles are the data shown in Fig 6E. The mean replication profile from two experiments is color-coded by strain and arranged by order of chromosome. The y-axis represents $T_{rep}$, the time at which 50% of cells in a population completes replication at a given genomic locus (See Materials and methods for detailed analysis). Rec8, recombination 8.
(TIF)

**S15 Fig. The budding index of yeast strains that are processed for replication profiling.**
Cells were arrested in G1 and then released as described in Figs 6E, S13 and S14. The y-axis

shows the fraction of budded cells in a population, measured as budding index. The mean (solid line) and standard deviation (shaded region) of two biological replicates for each strain are shown. Data associated with this figure can be found in S1 Data.
(TIF)

**S16 Fig. Deletion of genes encoding the Cdk8 complex alters genome replication and causes 8%–11% cost in wild type. (A)** The cell cycle progression profiles of *ssn2Δ*, *ssn3Δ*, or *ssn8Δ* strains compared to a wild-type control after release from a G1 arrest. (**B**) The fitness of *ssn2Δ*, *ssn3Δ*, or *ssn8Δ* strains relative to wild type, measured by competitive fitness assay. The darker gray points represent the values of three biological replicates and the thinner gray bar represents one standard deviation on each side of the mean of these measurements. The statistical significance between data from wild type and each mutant strain was calculated by two-tailed Student $t$ test, $**p < 0.01$. Data associated with this figure can be found in S1 Data. Cdk8, cyclin dependent kinase 8.
(TIF)

**S17 Fig. HU decreases the fitness difference between the Rec8-expressing strain and wild type.** The fitness of wild-type strains relative to that of the Rec8-expressing strain were measured in YPD and YPD containing 12.5 mM HU. The colored points represent the values of three biological replicates, and the darker gray bar represents one standard deviation on each side of the mean of these measurements (two-tailed Student $t$ test, $**p < 0.01$). Data associated with this figure can be found in S1 Data. HU, hydroxyurea; Rec8, recombination 8; YPD, yeast extract, peptone, and dextrose.
(TIF)

**S1 Table. Adaptive mutation fixed in each evolved population at generation 375.**
(PDF)

**S2 Table. Adaptive mutations fixed in each evolved population at generation 1,750.**
(PDF)

**S3 Table. Primers used for ChIP-qPCR.** ChIP, chromatin immunoprecipitation; qPCR, quantitative polymerase chain reaction.
(PDF)

**S1 Data. Original numerical values for Figs 1B, 1C, 2A, 2B, 2C, 2E, 3A, 3B, 3C, 3D, 4A, 4B, 4C, 4E, 5A, 5B, 5C, 5D, 5E, 5F, 6B, 6D, 6E; S1A, S1B, S2, S3, S6A, S8, S10, S12, S15, S16, and S17 Figs.**
(XLSX)

**S1 Raw Image. Original blot images for Figs 2A, 2B, S5A, S5B and S10 Figs.**
(PDF)

**S1 File. Non-synonymous mutations fixed in all the evolved populations at generations 375 and 1,750.**
(XLSX)

**S2 File. Strains used in this study.**
(XLSX)

**S3 File. Mutations that are present over 10% in all the evolved populations at generations 375 and 1,750.**
(XLSX)

## Acknowledgments

We thank Dana Branzei for providing yeast strain, Nichole Wespe and Marco Fumasoni for help with analysis of whole genome sequencing data, and Bauer Core Facility at Harvard for help with experiments. We thank Angelika Amon, Steve Bell, Jun-Yi Leu, Thomas LaBar, Andrea Giometto, Marco Fumasoni, and Hung-Ji Tsai for critical feedback on the manuscript. We thank members of the Murray lab and Marston lab for useful discussion.

## Author Contributions

**Conceptualization:** Yu-Ying Phoebe Hsieh, Andrew W. Murray.

**Data curation:** Yu-Ying Phoebe Hsieh.

**Formal analysis:** Yu-Ying Phoebe Hsieh, Daniel Robertson.

**Funding acquisition:** Adèle L. Marston, Andrew W. Murray.

**Investigation:** Yu-Ying Phoebe Hsieh, Vasso Makrantoni.

**Methodology:** Yu-Ying Phoebe Hsieh, Vasso Makrantoni, Daniel Robertson, Adèle L. Marston.

**Project administration:** Andrew W. Murray.

**Resources:** Andrew W. Murray.

**Software:** Daniel Robertson.

**Supervision:** Andrew W. Murray.

**Validation:** Yu-Ying Phoebe Hsieh, Vasso Makrantoni, Adèle L. Marston, Andrew W. Murray.

**Visualization:** Yu-Ying Phoebe Hsieh.

**Writing – original draft:** Yu-Ying Phoebe Hsieh, Andrew W. Murray.

**Writing – review & editing:** Yu-Ying Phoebe Hsieh, Vasso Makrantoni, Adèle L. Marston, Andrew W. Murray.

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
