## [Editor Report · Decision Letter 0]

5 Dec 2019

Dear Phoebe, 

Thank you for submitting your manuscript entitled "Evolutionary repair: changes in multiple functional modules allow meiotic cohesin to support mitosis" for consideration as a Research Article by PLOS Biology.

Your manuscript has now been evaluated by the PLOS Biology editorial staff, as well as by an academic editor with relevant expertise, and I'm writing to let you know that we would like to send your submission out for external peer review.

Please re-submit your manuscript within two working days, i.e. by Dec 09 2019 11:59PM.

Kind regards,

Roli

Senior Editor

PLOS Biology

---

## [Decision Letter · Decision Letter 1]

8 Jan 2020

Dear Phoebe,

Thank you very much for submitting your manuscript "Evolutionary repair: changes in multiple functional modules allow meiotic cohesin to support mitosis" for consideration as a Research Article by PLOS Biology. As with all papers reviewed by the journal, yours was evaluated by the PLOS Biology editors as well as by an Academic Editor with relevant expertise and in this case by three independent reviewers.

You'll see that all three reviews are rather positive, and the only request for additional analysis that I could see is one from reviewer #2 regarding an adjusted cut-off for fixation. Please also note the additional request from the Academic Editor below the reviewers' comments. Based on the reviews, we will probably accept this manuscript for publication, assuming that you will modify the manuscript to address the remaining points raised by the reviewers. Please also make sure to address the Data Policy and other policy-related requests noted at the end of this email.

We expect to receive your revised manuscript within two weeks. Your revisions should address the specific points made by each reviewer. In addition to the remaining revisions and before we will be able to formally accept your manuscript and consider it "in press", we also need to ensure that your article conforms to our guidelines. A member of our team will be in touch shortly with a set of requests. As we can't proceed until these requirements are met, your swift response will help prevent delays to publication.

*Copyediting*

*Published Peer Review History*

*Early Version*

*Submitting Your Revision*

Best wishes,

Roli

Senior Editor

PLOS Biology

DATA POLICY:

Many thanks for depositing the raw data in GEO and NCBI Bioprojects. Note that we also require that all individual quantitative observations that underlie the data summarized in the figures and results of your paper be made available in one of the following forms:

Regardless of the method selected, please ensure that you provide the individual numerical values that underlie the summary data displayed in the following figure panels as they are essential for readers to assess your analysis and to reproduce it: Figs 1BC, 2ABCE, 3BC, 4ABCE, 5ABCDEF, 6BDE, S1AB, S2, S3, S6, S8, S9, S10, S12, S15, S16B, S17. NOTE: the numerical data provided should include all replicates AND the way in which the plotted mean and errors were derived (it should not present only the mean/average values).

REVIEWERS' COMMENTS:

Reviewer #1:

This manuscript presents an unusual and interesting experiment. The authors test the consequences of and evolutionary response to replacing the mitotic alpha-kleisin with the diverged meiotic kleisin. First, they assay what the defects of expressing Rec8 instead of Scc1 are in their system (yeast cultures) - these include cohesion defects and earlier replication origin firing. They show that less cohesin associates with chromosomes and that cohesin is degraded faster by separase. The authors also show that the low expression of rec8 is not sufficient to account for ist lower accumulation on the chromosomes. Then they evolved 15 clones for 1750 generations and asked what the evolutionary response is to forcing cells to use REC8 in place of SCC1. They find that populations evolved mutations in several cohesin-interactors, the mediator complex, and later, mutations also in cell cycle regulators. They then functionally test these mutations and show that they do in fact improve growth relative to the ancestral strain. They show the adaptive mutations restore sister chromatid cohesion and delay replication origin firing.

I thought this study was really interesting, thorough, and the paper well written, and I have only minor comments:

Lines 205-206: when discussing the mitotic phases, please label these on the figures (i.e. on the y axis of Fig. 1D) so the uninitiated among us know what you are referring to.

Figure 1: Panel D - please label mitotic cell cycle phases on the y axis (or just refer to times in the text). Also - is there significance to the fact that the 180 minute peaks in scc1- and to a lesser extent in the Rec8 line are fatter and wider than those of wild type? Is replication timed more messily? Is there aneuploidy? Also, can you rule out that the bigger 2C peak is not due to polyploidy?

Fig. S6 is very helpful and might be good to incorporate into Fig. 3.

By generation 1750 several of the aneuploids for Chr 9 had lost the extra chromosome. Did they gain a local duplication or change in expression of SCC3, or did the additional mutations that evolved obviate the extra SCC3?

Line 40: «direct and direct» - you probably mean direct and indirect?

Line 47: Perhaps «answering this question» instead of «this problems»

Line 245: «In the bar graph, the upper arrow marks…» - I think you mean in the blot.

Reviewer #2:

[identifies himself as Andreas Hochwagen]

This study investigates the evolutionary cost associated with the sub-functionalization of the cohesin complex in budding yeast. Mitotic and meiotic yeast use cohesin complexes that differ in their kleisin subunits. Previous work had found that expression of the meiotic kleisin, REC8, in place of mitotic kleisin, SCC1, leads to fitness defects. The authors now show that this defect is linked to incomplete sister chromatid cohesion, leading to increased rates of chromosome misalignment and premature loss of cohesion. They also observe an unexpected acceleration of DNA replication timing in mitotic cells expressing REC8. Evolution experiments identified a number of spontaneous suppressor mutations of these defects. Several of the suppressor mutations affect the mediator complex, the cell cycle entry machinery or cohesin regulation, but no mutations were observed in REC8. Intriguingly, all of these suppressors appear to rescue the early S phase phenotype of REC8 misexpression. 

This is a very nice and interesting paper. The writing is clear and the experiments are generally conclusive. I only have a few minor comments.

1. I wonder whether the accelerated S phase of REC8-expressing cells could be related to defects in recruiting Rad53. The Longhese group showed a number of years ago that meiotic chromosomes do not activate Rad53. Since Rad53 is an important factor in regulating the timing of origin firing, a failure to properly activate Rad53 may lead to accelerated S phase. Exploring this possibility is obviously beyond the scope of this paper, but it may be worth considering this possibility in the discussion.

2. One thing that needs clarification is the mutation calling for REC8. The authors speculate that the absence of mutations in REC8 reflects differences in target size between the observed suppressors and REC8. However, the way the experimental strain is constructed, REC8 is present in two copies: one on the plasmid under the SCC1 promoter, and one in the genome under its own promoter. As a result there will always be a 50% background from the unexpressed genomic REC8 copy in the sequencing reads. The authors focus only on fixed mutations, and define fixed mutations as mutations found in >90% of reads. This cutoff would make it impossible for rec8 mutations to reach fixation given the expected 50% wild-type sequences from the genomic REC8 copy. This should be clarified and probably warrants reanalysis of the REC8 sequences with a different cutoff.

3. Were any partial aneuploidies observed or were all aneuploidies affecting entire chromosomes?

4. Line 537: I am not sure I understand the argument put forward about the relative origin firing. I agree that earlier origins generally have a lesser chance of being inactivated by nearby origins, but the authors also show that the temporal order of origin firing does not seem to be affected. Doesn't that mean that speeding up replication for any particular origin is irrelevant because all the neighboring origins will also speed up proportionally? At least based on this I would assume that there is no increased chance of avoiding inactivation by a neighboring origin.

Typos:

Line 122: Scc1-expressing

Line 328: both CLN2 and SWI4 mutated

Line 564: which delays replication origin firing

Reviewer #3:

Hsieh and colleagues use experimental evolution to examine if and how cells affected in mitotic cohesion function are able to restore fitness. They find that replacing the mitotic kleisin gene SCC1 with its meiosis-specific paralog REC8 results in a fitness decrease due to reduced cohesin activity. Evolving 15 parallel mutant populations for 1750 generations partly restores fitness through a combination of mutations located in in loci related to the mediator complex (Cdk8 module), cohesin function (ESP1, SMC1 and SMC3) and cell cycle regulation (MBP1, CLN2). Interestingly, none of the compensatory mutations were located in or near the REC8 gene, suggesting that the fitness effect of single mutations in REC8 is likely limited, while the frequency of combinations of mutations in REC8 that may confer stronger fitness benefits, is too low.

This is an exceptionally eloquent paper that draws from an impressive body of work. Perhaps the biggest concern is that the study does not bring much truly novel biological insight. That said, the paper does yield two major conclusions. Firstly, it helps to shed light on the biological processes controlling cohesin activity. Specifically, the results show how proper cohesin function does not only require proper assembly of the cohesion complex, but also a sufficiently slow progression through S phase as well as some -as yet unknown- mediator activity (nicely summarized in Fig 8). Second, the results remind us how evolving suppressor phenotypes often depends on mutations that occur relatively frequently, such as structural variation and (partial) loss-of-function mutations. This may be especially the case in laboratory conditions, where such relatively "blunt" evolutionary solutions may not have the same pleiotropic (negative) fitness consequences as they would have in more complex natural settings. Perhaps this second point should be stressed even more in the discussion section.

Minor remarks

I found Figure 1 panel A to be a bit cluttered and unclear; pls consider refining

Figure 3B - would it perhaps make more sense to compare everything to the WT fitness level? Also, pity that the WT was not evolved in parallel with the REC8 mutant.

The text is a bit long and therefore loses a bit of its poise and punch.

REQUEST FROM THE ACADEMIC EDITOR:

[I would like to see] a more nuanced discussion of paralog divergence (e.g., what would the ancestral situation have been? what are the effects of the adaptive mutations in the context of normal Scc1 or in the context of Rec8 functioning during meiosis?).

---

## [Editor Report · Decision Letter 2]

21 Feb 2020

Dear Dr Hsieh,

On behalf of my colleagues and the Academic Editor, Mark L Siegal, I am pleased to inform you that we will be delighted to publish your Research Article in PLOS Biology. 

Early Version

PRESS 

Kind regards,

Vita Usova

Publication Assistant, 

PLOS Biology

on behalf of

Roland Roberts,

Senior Editor

PLOS Biology